# Beyond Data Filtering: Knowledge Localization for Capability Removal in LLMs

## Abstract

Large Language Models increasingly possess capabilities that carry dual-use risks. While data filtering has emerged as a pretraining-time mitigation, it faces significant challenges: labeling whether data is harmful is expensive at scale, and given improving sample efficiency with larger models, even small amounts of mislabeled content could give rise to dangerous capabilities. To address risks associated with mislabeled harmful content, prior work proposed Gradient Routing (Cloud et al., 2024) – a technique that localizes target knowledge into a dedicated subset of model parameters so they can later be removed. We explore an improved variant of Gradient Routing, which we call Selective GradienT Masking (SGTM), with particular focus on evaluating its robustness to label noise. SGTM zero-masks selected gradients such that target domain examples only update their dedicated parameters. We test SGTM's effectiveness in two applications: removing knowledge of a language from a model trained on a bilingual synthetic dataset, and removing biology knowledge from a model trained on English Wikipedia. In both cases SGTM provides better retain/forget trade-off in the presence of labeling errors compared to both data filtering and a previously proposed instantiation of Gradient Routing. Unlike shallow unlearning approaches that can be quickly undone through fine-tuning, SGTM exhibits strong robustness to adversarial fine-tuning, requiring seven times more fine-tuning steps to reach baseline performance on the forget set compared to a traditional unlearning method (RMU). Our results suggest SGTM provides a promising pretraining-time complement to existing safety mitigations, particularly in settings where label noise is unavoidable.

## 1 Introduction

As LLMs grow more capable, concerns are being raised over their potential misuse – ranging from software exploits to dangerous chemical, biological, radiological, and nuclear (CBRN) applications (Urbina et al., 2022; Kang et al., 2024). Post-training mitigations, such as refusal training or output classifiers, are improving, yet continue to face challenges from determined adversaries (Andriushchenko & Flammarion, 2024; McKenzie et al., 2025). This motivates interventions earlier in the training pipeline, to prevent models from acquiring certain capabilities in the first place.

A common pretraining-time approach, data filtering aims to exclude harmful or restricted content before it can be learned (O'Brien et al., 2025; Chen et al., 2025; Maini et al., 2025). Achieving comprehensive and precise filtering at scale is challenging: acquiring high-quality labels is expensive at scale (Anwar et al., 2024), undesired content is often embedded within benign documents (Dodge et al., 2021), and many concepts are entangled between harmful and beneficial use cases (Pannu et al., 2025). This leads to an inevitable trade-off: developers must either accept false negatives (retaining dangerous content), or remove data useful for general capabilities (O'Brien et al., 2025).

Recent research proposed localizing target knowledge to a subset of the model's parameters, which can later be erased to remove knowledge from the model. Methods include Gradient Routing (Cloud et al., 2024), which achieves localization by modifying gradients during pretraining, and Redirection for Erasing Memory (Schoepf et al., 2025), applied post-training. Both methods outperform data filtering in terms of removal performance in the presence of labeling errors.

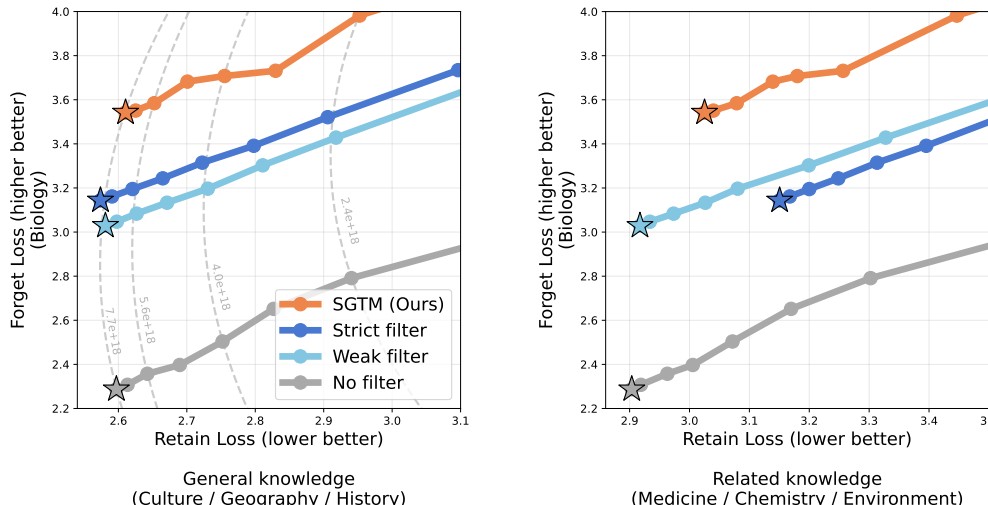

Figure 1: **Retain/forget trade-off when removing biology knowledge from a model trained on Wikipedia.** We compare Selective GradienT Masking (**SGTM**) with two data filtering strategies: **weak** (removing biology category only) and **strict** (also removing medicine, chemistry and environment categories). The goal is to remove biology knowledge from the model. Each line represents the progress of a training run, each point a checkpoint at equal intervals. Stars show final checkpoints. Dashed lines show equal compute expenditure in FLOPs (not shown on right). On general knowledge (left) and biology-adjacent knowledge (right) SGTM provides superior retain/forget trade-off – higher forget loss at any given retain loss value. SGTM incurs a compute efficiency penalty, showing higher loss on general knowledge at the end of training compared to both filtering strategies.

We explore an improved variant of Gradient Routing, which we call **Selective GradienT Masking (SGTM)**. SGTM first designates a portion of the model weights to be dedicated to a predetermined domain (e.g. CBRN), allocating certain MLP neurons and attention heads in each transformer block. During training, it selectively zero-masks gradients from examples representing the target domain such that they only update the dedicated portion of the network. After training, it removes the undesired capabilities by zeroing out the dedicated portion of the network, leaving the rest of the model's knowledge mostly intact. Compared to the Gradient Routing variant originally proposed by Cloud et al. (2024), which masks activation gradients on a subset of layers, SGTM's parameter-gradient masking is less disruptive to retain performance and, when applied across all layers, more effectively restricts information flow from forget data into non-forget parameters (See Appendix B).

We first show SGTM's robustness to label noise in a clean synthetic setup with access to ground truth labels, removing the knowledge of a language from a model trained on a bilingual dataset. We show that SGTM outperforms the original Gradient Routing variant on both retain and forget data (Figure 3), and provides better retain/forget trade-off in the presence of labeling errors compared to data filtering (Figure 4, left). We also quantify the rate of data leakage from forget data into non-foget parameters across model sizes, finding that it decreases as models grow larger (Figure 5).

We then apply SGTM to a realistic large-scale dataset, training a 254M parameter model on English Wikipedia, targeting biology knowledge for removal, demonstrating SGTM's performance under realistic label noise from a content classifier. SGTM provides better retain/forget trade-off than both weak (removing only biology data) and strict (also removing medicine, chemistry and environment data) filtering baselines (Figure 1). We show SGTM to be the best performing Gradient Routing variant on this task (Figure 7). Evaluated by compute efficiency, SGTM slows training on general knowledge by 6% (Figure 9).

Finally, we show SGTM to be robust to adversarial fine-tuning (Figure 4, right), with relearning speed much slower than Representation Misdirection for Unlearning (RMU) (Li et al., 2024), a state-of-the-art traditional unlearning technique. It takes SGTM 7× more forget tokens in fine-tuning than RMU to achieve the baseline forget loss (92M vs 13M tokens). In the same setup, weak data filtering takes 85M and strict data filtering takes 92M forget tokens to achieve the baseline loss.

## 2 RELATED WORK

**Post-training safety mitigations.** A common line of defense is to apply mitigations after pretraining. Refusal training teaches models to decline unsafe requests, but these safeguards can often be bypassed through jailbreaks and prompt engineering (Kumar et al., 2024; Andriushchenko & Flammarion, 2024). Output classifiers – auxiliary models that filter generated text – can be circumvented by determined adversaries (Schwinn et al., 2023; McKenzie et al., 2025). Machine unlearning techniques instead attempt to erase specific knowledge from trained models (Yao et al., 2024; Liu et al., 2024; Barez et al., 2025; Liu et al., 2025), but remain brittle: suppressed information can often be recovered through adversarial fine-tuning (Deeb & Roger, 2024; Lermen et al., 2023), benign fine-tuning on unrelated tasks (Hu et al., 2024), jailbreaks (Łucki et al., 2024), or rephrased queries (Lynch et al., 2024).

**Pre-training safety mitigations.** Pre-training data filtering is increasingly adopted by frontier model developers (OpenAI, 2025; Meta, 2025; Anthropic, 2025; Agarwal et al., 2025; Kamath et al., 2025) and is effective at improving model safety (Maini et al., 2025; Li et al., 2025). By preventing the initial acquisition of dangerous knowledge, data filtering proves to be more robust to fine-tuning attacks than post-hoc unlearning (O'Brien et al., 2025). However, data filtering faces a critical challenge: acquiring high-quality labels at scale. The enormous size of pre-training datasets forces developers to rely on cheap, imperfect filtering strategies such as keyword filters, heuristics, and lightweight classifiers (Longpre et al., 2024; Stranisci & Hardmeier, 2025; Chen et al., 2025; Albalak et al., 2024). These approaches suffer from high false positive rates and miss nuanced harmful content requiring contextual understanding (Welbl et al., 2021; Paullada et al., 2021). For instance, the hazardous biology classifier proposed by O'Brien et al. (2025) achieves only 44% precision at 98% recall, leading to the removal of over 8% of training data.

Beyond filtering, several works modify the pre-training objective to reduce harmful generation. Welleck et al. (2020) and Korbak et al. (2023) condition the next-token loss on reward-model scores, shifting the generation distribution away from undesirable outputs. Wang et al. (2025) masks or penalizes loss on high-risk tokens, explicitly encouraging models to retain understanding of such content while suppressing its generation. In contrast, rather than modifying generation behavior, our approach prevents the underlying knowledge from being learned at all.

**Knowledge Localization.** Recent work has explored an alternative pretraining-time approach to localize specific knowledge to particular model parameters during training, enabling targeted removal. Inspired by modular architectures that separate knowledge across specialized components (e.g., Mixture of Experts (Shazeer et al., 2017; Gururangan et al., 2021; Park et al., 2025), modular architectures (Jacobs et al., 1991a;b; Andreas et al., 2016; Alet et al., 2018; Kirsch et al., 2018; Ruder et al., 2019; Pfeiffer et al., 2023), or adapters (Hu et al., 2022; Ponti et al., 2023)), these methods explicitly enforce localization through gradient control to allow strict localization of specific knowledge that one might wish to remove.

The gradient masking in SGTM mirrors adapter methods like LoRA (Hu et al., 2022): while the forward pass uses all model parameters, the backward pass selectively updates only forget-specific parameters, analogous to how LoRA restricts gradient updates to adapter modules while keeping the base model frozen.

Cloud et al. (2024) propose Gradient Routing, applying weighted, data-dependent masks to the model's computation graph to localize harmful knowledge into a designated subset of weights. Ghosal et al. (2025) apply a similar approach focused on MLP layers to localize memorization of specific examples. Schoepf et al. (2025) iteratively localizes undesired knowledge to newly added neurons post-training. These methods share a crucial advantage over data filtering: the *absorption* property (Cloud et al., 2024). Even when some harmful examples are mislabeled as benign, gradient routing mechanisms can partially localize their impact to the designated parameters, maintaining effective removal despite labeling errors. Both Cloud et al. (2024) and Schoepf et al. (2025) demonstrate robustness to discovery rates as low as 50% of harmful samples, a scenario where data filtering fails. Our proposed method, SGTM, further improves the trade-off between retaining general capabilities and removing target knowledge, achieving better retain/forget trade-offs while maintaining robustness to labeling errors.

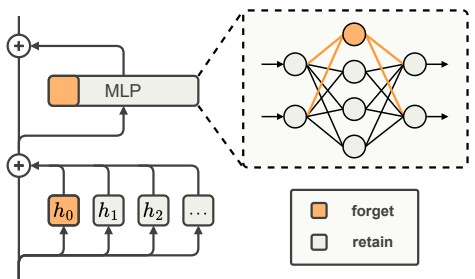

Figure 2: **Forget/Retain parameter split in Selective Gradient Masking.** In each transformer block we designate certain number of attention heads and MLP hidden units to the forget data (orange). The remaining parameters are designated to the retain data.

| Data | Intervention | | Parameters updated | |
| | Forward pass | Backward pass | $\theta_{\text{forget}}$ | $\theta_{\text{retain}}$ |
|---|---|---|---|---|
| $\mathbf{D}_{\text{forget}}$ | — | Mask retain gradients $(\nabla_{\theta_{\text{retain}}} = 0)$ | ✔ | ✘ |
| $\mathbf{D}_{\text{unlabeled}}$ | — | — | ✔ | ✔ |
| $\mathbf{D}_{\text{retain}}$ | Mask forget parameters $(\theta_{\text{forget}} = 0)$ | — | ✘[1] | ✔ |

[1] Due to the associated activations being set to zero.

Table 1: **Training interventions applied to different data subsets in SGTM.** Interventions are described in Section 3.2. Empty intervention (—) indicates that normal training procedure is followed.

## 3 METHOD

### 3.1 NOTATION

We consider a transformer block (Vaswani et al., 2017) consisting of a multi-head attention and an MLP layer, with $h$ attention heads, model dimension $d$, and MLP dimension $d_{\text{MLP}}$. We designate $h_{\text{forget}}$ (out of $h$) attention heads and $d_{\text{forget}}$ (out of $d_{\text{MLP}}$) MLP hidden units for the forget data, and the remaining attention heads and MLP units for the retain data. We split parameters in the relevant linear layers into forget and retain segments across all transformer blocks. Figure 2 provides a simplified visualization. We provide a detailed explanation of parameter designation in Appendix F.

We refer to $\theta_{\text{forget}}$ and $\theta_{\text{retain}}$ to mean all parameters in the model with the given designation. We can write the set of all model's parameters as $\theta = \{\theta_{\text{forget}}, \theta_{\text{retain}}\}$. Parameters outside transformer blocks (namely, embeddings) are considered part of $\theta_{\text{retain}}$, unless explicitly specified otherwise.

For the training data, we denote forget and retain data distributions as $\mathcal{D}_{\text{forget}}$ and $\mathcal{D}_{\text{retain}}$ respectively. Our goal is to train a model that performs well on $\mathcal{D}_{\text{retain}}$, but poorly on $\mathcal{D}_{\text{forget}}$. Note that these are idealized oracle data distributions and might not be accessible in practice. We then refer to the actual training dataset as $\mathbf{D}$. Accounting for the realistic data labeling, we assume $\mathbf{D}$ to be split into three subsets: $\mathbf{D} = \{\mathbf{D}_{\text{forget}}, \mathbf{D}_{\text{retain}}, \mathbf{D}_{\text{unlabeled}}\}$. $\mathbf{D}_{\text{forget}}$ and $\mathbf{D}_{\text{retain}}$ are intended to contain samples where the input classifier is confident in the corresponding label, while uncertain or ambiguous samples would be a part of $\mathbf{D}_{\text{unlabeled}}$.

### 3.2 TRAINING INTERVENTIONS

Our method performs two types of interventions during training as summarized in Table 1.

**Selective Gradient Masking.** For samples from $\mathbf{D}_{\text{forget}}$, we apply selective gradient masking during the backward pass so that these samples do not update $\theta_{\text{retain}}$. We first compute gradients for all parameters normally, and then zero out gradients for $\theta_{\text{retain}}$ before applying the optimizer ($\nabla_{\theta} = \{\nabla_{\theta_{\text{forget}}}, 0\}$). Masking parameter gradients rather than activation gradients is the key distinction from the prior Gradient Routing method (Cloud et al., 2024). While both approaches prevent updates to $\theta_{\text{retain}}$ on forget examples, masking activation gradients is more disruptive, because it blocks backpropagation through the masked activations, altering gradients for all remaining parameters. It also permits greater information flow from $\mathbf{D}_{\text{forget}}$ into non-forget parameters, since activation-gradient masking does not block updates to down-projection layers (See Appendix B).

**Selective Parameter Masking.** For samples from $\mathbf{D}_{\text{retain}}$ we apply selective parameter masking during the forward pass to train the model to perform well on $\mathcal{D}_{\text{retain}}$ even when $\theta_{\text{forget}}$ parameters are set to 0. In particular, we zero-mask $\theta_{\text{forget}}$ parameters during the forward pass, leading to corresponding activations being set to zero as well.

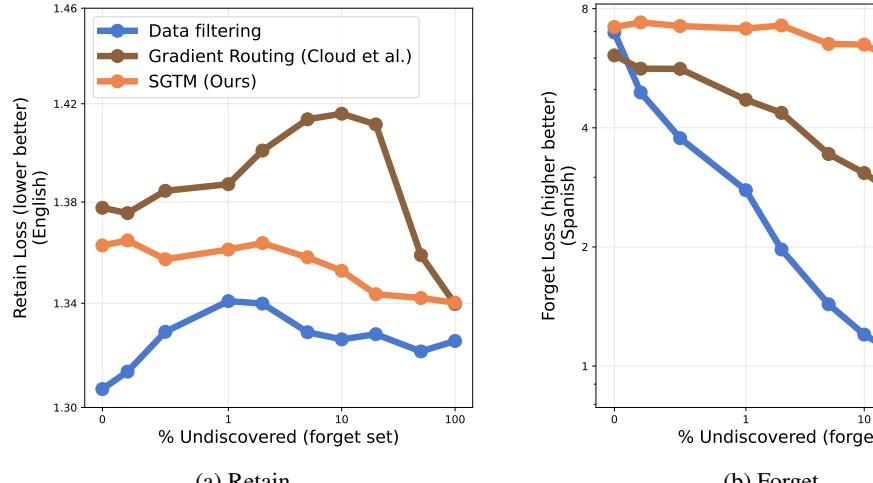

(a) Retain

(b) Forget

Figure 3: **SGTM robustly removes forget knowledge, remaining effective even when large fractions of forget data are unlabeled.** We report calibrated losses on (a) retain and (b) forget sets when attempting to remove Spanish from a model trained on bilingual (English/Spanish) TinyStories dataset. We vary the percentage of undiscovered forget data, *i.e.,* the proportion of the forget set not labeled as such. (a) SGTM consistently has lower retain loss than Gradient Routing (Cloud et al., 2024), but higher retain loss than data filtering. (b) For all non-zero labeling error rates considered, SGTM demonstrates stronger forgetting than both Gradient Routing and data filtering.

**Ablation.** After the training is complete, we set $\theta_{\text{forget}} = 0$ to remove knowledge specific to $\mathcal{D}_{\text{forget}}$.

## 4 SYNTHETIC DATASET (TINYSTORIES) RESULTS

### 4.1 EXPERIMENTAL SETUP

In this setup we aim at localizing and removing the knowledge of one language from a model trained on a bilingual dataset, with a varying level of label noise. We train a 64M-parameter model on 1.2B tokens from bilingual TinyStories dataset (Eldan & Li, 2023) in English and Spanish for 1 full epoch, roughly following Chinchilla-optimal scaling (Hoffmann et al., 2022). We acquire the Spanish version by translating the original English dataset with Claude 3 Haiku.

We treat English data as retain ($\mathcal{D}_{\text{retain}}$) and Spanish data as forget ($\mathcal{D}_{\text{forget}}$). When constructing datasets $\{\mathbf{D}_{\text{forget}}, \mathbf{D}_{\text{retain}}, \mathbf{D}_{\text{unlabeled}}\}$ under the perfect labeling, all of the Spanish data is allocated to $\mathbf{D}_{\text{forget}}$, and English data is randomly split between $\mathbf{D}_{\text{unlabeled}}$ (75%) and $\mathbf{D}_{\text{retain}}$ (25%).

Note that in this setup we have access to the ground truth labels, as the training data is synthetically generated. To quantify robustness to labeling errors, we introduce artificial mislabeling. We define "undiscovered forget percentage" as the percentage of all Spanish data that is allocated to $\mathbf{D}_{\text{unlabeled}}$ instead of $\mathbf{D}_{\text{forget}}$ (*i.e.,* not explicitly labeled as forget data). This could also be seen as FNR (False Negative Rate) of the hypothetical classifier identifying the forget data.

As we are approximating a more realistic scenario when only a small portion of the weights would be dedicated to the forget knowledge, we designate 1 (out of 32) attention heads and 64 (out of 2048) MLP hidden units as $\theta_{\text{forget}}$, with the remaining parameters designated as $\theta_{\text{retain}}$. In this scenario $\mathcal{D}_{\text{forget}}$ has a lot of unique tokens not present in $\mathcal{D}_{\text{retain}}$, so we update embeddings with both forget and retain data. For training with data filtering we simply ignore $\mathbf{D}_{\text{forget}}$ and train for one epoch on $\{\mathbf{D}_{\text{unlabeled}}, \mathbf{D}_{\text{retain}}\}$.

All losses reported in this section are calibrated over forget (Spanish) and retain (English) datasets. Calibration is computed with a trained logit bias post-training (see Appendix I). This calibration avoids superficially high losses post-ablation due to extremely low probabilities of relevant tokens. Further details on the training process, logit calibration and the dataset are presented in Appendix I.

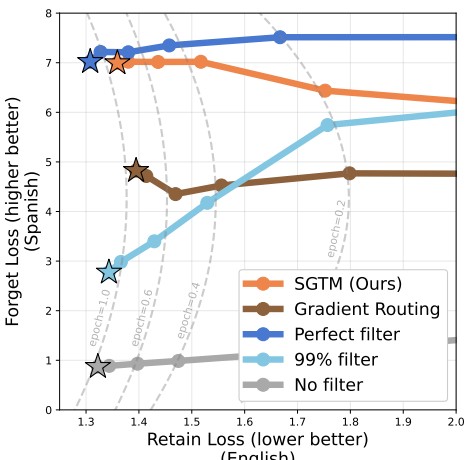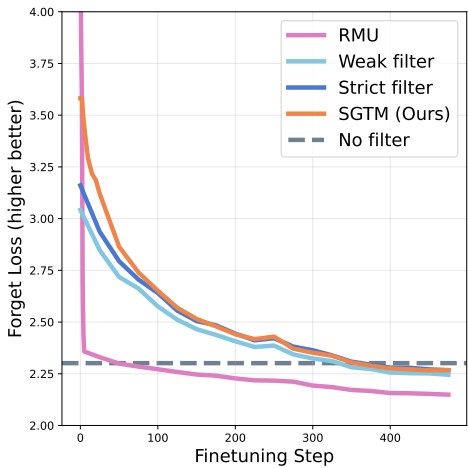

Figure 4: **Retain/forget trade-off when removing the knowledge of a language from a bilingual model (left).** We show the trade-off between forget and retain loss on the task of removing Spanish knowledge from the bilingual TinyStories model. We set the rate of undiscovered forget data to 1%. Each line represents the progress of one training run, and each point is a checkpoint at equal intervals of the training. Stars show the final checkpoint. Dashed lines show the same proportion of training completed. We compare SGTM with data filtering (removing 99% of data) and Gradient Routing (Cloud et al., 2024). We also show "perfect filter" and "no filter" training as a reference. SGTM provides a better trade-off (higher forget loss at any fixed value of retain loss) than both 99% filter and Gradient Routing, closely approximating the oracle model represented by perfect filtering. **SGTM's knowledge removal from the Wikipedia model is robust to adversarial fine-tuning (right).** We measure the relearning rate by performing adversarial fine-tuning after removing biology knowledge from a model trained on Wikipedia. RMU (a state-of-the-art traditional post-training machine unlearning method) is brittle, quickly reaching the baseline forget loss in only 50 steps. SGTM (350 steps) is as robust as strict data filtering (350 steps), narrowly outperforming weak filtering (325 steps). It also maintains an advantage over strict data filtering for the first 150 fine-tuning steps. Each fine-tuning step represents 260k forget tokens.

## 4.2 RESULTS

Figure 3 shows retain (a) and forget (b) loss with increasing rates of undiscovered forget data. SGTM maintains strictly better performance – lower retain loss and higher forget loss – than previously proposed variant of Gradient Routing (Cloud et al., 2024) across all discovery rates tested. For data filtering, forget loss drops quickly when even a few forget samples are not filtered out. Both knowledge localization methods (SGTM and Gradient Routing) show slower decline in forget loss compared to data filtering as the rate of undiscovered forget data increases. On the retain set, however, SGTM shows higher loss than data filtering.

A natural concern would be that SGTM's higher forget loss simply reflects a generally degraded model compared to data filtering, rather than successful forgetting. To explore this possibility, Figure 4 shows the trade-off between forget and retain performance throughout training for SGTM, Gradient Routing, and three filtering options: perfect filter (0% undiscovered), 99% filter (1% undiscovered) and no filter (100% undiscovered). Though SGTM has slightly higher final retain loss than 99% filter (roughly equivalent to the 99% filter's checkpoint at 80% of training), it offers better forget-retain trade-off – SGTM has higher forget loss at any fixed retain loss value. Note that SGTM aims to remove knowledge learned from forget training data as if was never seen by the model, and as such we do not expect or aim for it to outperform perfect data filtering. However, SGTM closely approximates forgetting performance of the perfect filter, with almost equivalent forget loss at the end of the training.

Here we've considered type II errors of the data classifier (false negative forget samples). We perform a detailed analysis with a full range of false positive and false negative rates in Appendix E.

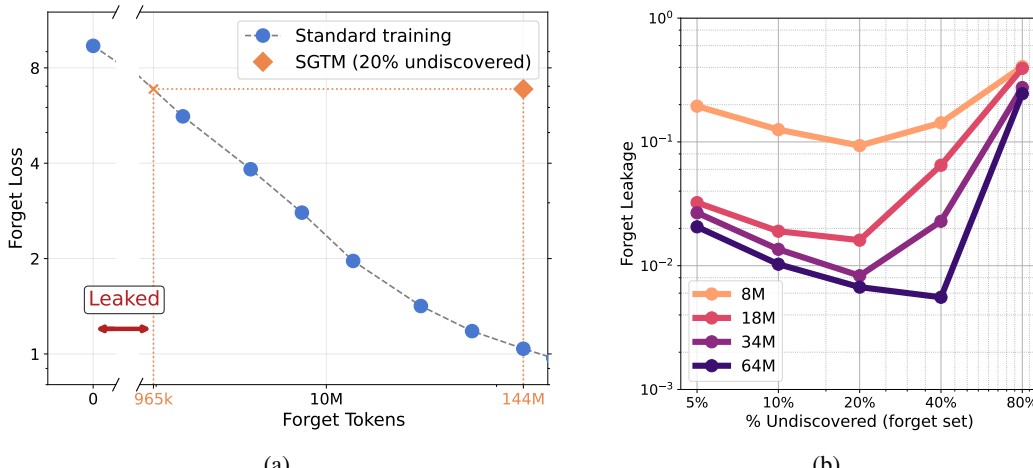

(a)                 (b)

Figure 5: **(a) Leakage is quantified via equivalent standard training comparison with variable number of forget tokens added to the data mix.** The baseline curve (blue) maps the relationship between forget token exposure and forget loss established by training models on all retain data with increasing amounts of forget tokens added. Each blue point represents a model trained with standard training procedure with a given number of forget tokens added to the training dataset. For a given SGTM run (orange) we then take its forget loss and find the number of forget tokens that would achieve the same loss when added to the data mix in standard training (965k). The leakage is then computed by normalizing this number by the total number of (unlabeled) forget tokens in SGTM run. **(b) Leakage decreases with model scale.** Values denote the ratio of leaked information (measured in forget token exposure) to total undiscovered forget tokens, ranging between 0 (no leakage) and 1 (all information leaked). Larger models consistently exhibit lower leakage rates, with the 64M model maintaining leakage below 0.02 for up to 40% undiscovered forget data.

### 4.3 LEAKAGE

In Section 4.2, we demonstrate that SGTM maintains high forget loss even when unlabeled portion of the training data ($\mathbf{D}_{\text{unlabeled}}$) contains forget examples. In this section, we quantify how much information from these undiscovered samples leaks into the retain parameters ($\theta_{\text{retain}}$) using a metric we call *leakage*. Here we provide an intuitive definition of the metric – see Appendix G for a formal definition. To measure leakage, we first establish a baseline relationship between forget token exposure and model performance, as shown in Figure 5(a). We train standard models (*i.e.,* without SGTM) on the complete retain dataset combined with increasing rates of forget tokens added to the training data. This creates a mapping between the number of forget tokens in training data and resulting forget loss, while keeping retain data constant to control "general" capabilities. Note that in this section we report losses without logit calibration to reduce the number of hyperparameters and simplify the comparison across model sizes.

With this baseline, we convert forget loss values to equivalent forget token exposure. For example, an SGTM model trained with 20% undiscovered forget data (144M tokens) achieves forget loss equivalent to a baseline model trained on 965k forget tokens. This means that with 144M undiscovered forget tokens seen by the model, the model gained as much information on the target domain as a standard model would from 965k forget tokens. Note that SGTM is always trained on full retain and forget datasets, and undiscovered forget tokens here refer to those not labeled as such, *i.e.,* where masking is not applied.

We define this ratio of the equivalent tokens to the total undiscovered tokens as *leakage*. Note that under this definition, the leakage rate of data filtering would always be 1, as data filtering model with a certain percentage of undiscovered forget tokens is equivalent to a standard model with all undiscovered forget tokens in the training data.

We evaluate leakage across a range of model sizes (8M to 64M parameters), scaling dataset size following Chinchilla-optimal principles (detailed training configurations are provided in Appendix I.1).

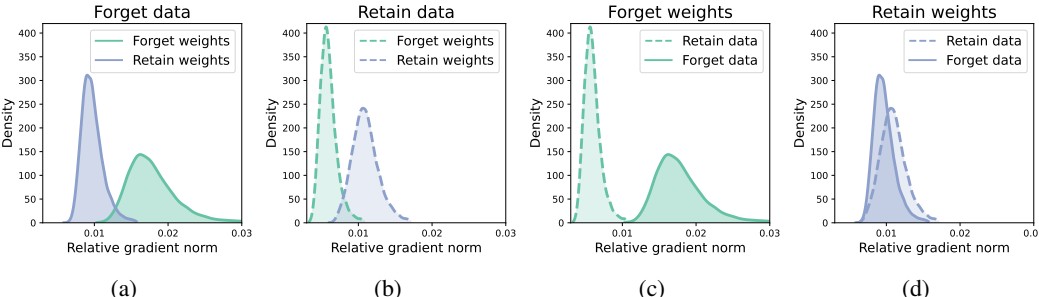

Figure 6: **Unlabeled forget data mostly update forget parameters, and unlabeled retain data mostly update retain parameters.** Each panel shows kernel density estimates of relative gradient norms ($|\nabla_\theta|/|\theta|$) for different parameter-data combinations. Gradients are computed for unlabeled data, i.e. no masking is applied. Forget parameters (green) and retain parameters (blue) are evaluated on both forget data (solid) and retain data (dashed) from the test set, with no gradient masking applied. Forget data predominantly updates forget parameters (a), while retain data predominantly updates retain parameters (b). Conversely, forget parameters receive much stronger updates from forget data (c). Retain parameters receive updates of similar magnitude from either forget and retain data, with slightly stronger updates from the retain data (d).

We maintain fixed forget dimensions ($d_{\text{forget}} = 64$, $h_{\text{forget}} = 1$), which means that the proportion of forget-designated parameters naturally decreases with model scale.

Figure 5(b) shows leakage values across a range of undiscovered forget rate values for models of varying size. First, for the largest model – 64M parameters, trained for one full epoch on the bilingual TinyStories dataset in a near Chinchilla-optimal configuration – leakage remains remarkably low: between 0.005 and 0.02 for undiscovered forget rates up to 40%.

Second, we observe a clear inverse relationship between model scale and leakage rates. Across all levels of undiscovered forget data, larger models consistently exhibit lower leakage than their smaller counterparts. This scaling behavior is particularly encouraging for the application of SGTM to larger-scale models, suggesting that the method's effectiveness improves as models grow larger. This trend contrasts with data filtering, where recent work on data poisoning (Souly et al., 2025) shows that the number of malicious or mislabeled samples required to influence model behavior remains roughly constant with scale – implying that larger models demand increasingly accurate classifiers to maintain the same level of protection.

## 4.4 GRADIENT NORMS

To understand SGTM's robustness to label noise, we hypothesize that the model develops self-reinforcing knowledge localization: once labeled forget examples begin steering updates into $\theta_{\text{forget}}$, unlabeled forget samples naturally follow the same pathway and send stronger gradient signals to forget parameters even without explicit masking. To evaluate this, we compute relative gradient norms ($|\nabla_\theta|/|\theta|$) in a model trained under perfect labeling, and then measure per-sample gradients for forget and retain test examples while treating all samples as $\mathbf{D}_{\text{unlabeled}}$ (i.e., without any masking).

Figure 6 shows that unlabeled forget data predominantly updates forget parameters, and unlabeled retain data predominantly updates retain parameters, confirming specialization. Crucially, unlabeled forget samples produce substantially larger updates to $\theta_{\text{forget}}$ than unlabeled retain samples, providing direct evidence for the *absorption* mechanism conjectured by Cloud et al. (2024), and explaining why SGTM remains robust when labels are imperfect: misclassified forget samples still preferentially reinforce the forget-designated subnetwork rather than leaking into $\theta_{\text{retain}}$. Interestingly, retain parameters exhibit a much weaker separation, suggesting that magnitude alone does not fully characterize knowledge transfer.

## 5 REALISTIC DATASET (WIKIPEDIA) RESULTS

### 5.1 EXPERIMENTAL SETUP

In this setup we aim at localizing and removing the biology knowledge from a 254M parameter model trained on 3.7B tokens from an English Wikipedia (Wikimedia). We label each Wikipedia article with an `articletopic` property provided by Wikipedia's content classification service (Halfaker & Geiger, 2020; Wikimedia, 2025). The taxonomy is hierarchical, covering 64 topics grouped into high-level categories: Culture, Geography, History and Society, and STEM (see Table 5 for the full taxonomy). We apply one category per article, choosing the one with highest classifier score.

We then consider the "STEM.Biology" subcategory (3.7% of overall tokens) as forget set, and all remaining data as retain set. For evaluation we consider 3 separate test sets: **forget** (Biology), **related knowledge** (Earth and Environment, Medicine and Health, Chemistry) and **general knowledge** (Culture, Geography, History and Society). In each subcategory we set aside 5000 articles for the test set. For general and related knowledge we report average loss over test sets for all subcategories. For training, $D_{forget}$ contains all data labeled as biology, while remaining non-biology data is randomly split between $D_{unlabeled}$ (90%) and $D_{retain}$ (10%).

We consider two data filtering strategies. "Weak filter" only removes data labeled as biology from the training set. "Strong filter" additionally removes related categories, which are most likely to contain biology knowledge (Earth and Environment, Medicine and Health, Chemistry). With both data filtering strategies we maintain the same overall compute budget, training for the same number of steps on the remaining data after filtering.

When training with SGTM, we designate $h_{forget} = 1$ (out of 32) attention head and $d_{forget} = 64$ (out of 4096) MLP hidden units as $\theta_{forget}$. As in Section 4, here we report the loss after calibration with a trained logit bias post-training to avoid superficially high loss values post-ablation. Reported losses are averaged over 3 independent runs. We provide further details on the training process, logit calibration and the dataset in Appendix J.

### 5.2 RESULTS

We first note that here, unlike the synthetic data setup from Section 4, we do not introduce additional labeling errors, and demonstrate the effectiveness of SGTM under more realistic conditions with natural label noise. With document-level labeling we expect that documents not labeled as biology could also contain biology knowledge, and an algorithm with strong label noise robustness should outperform strict filtering in this scenario.

Figure 1 (left) shows the trade-off between the performance on the forget set and on general knowledge. Controlling for retain loss, SGTM achieves a higher forget loss than both filtering methods. Among filtering methods, strict filter shows stronger forgetting, as it removes more biology-adjacent data from the training set. Both filtering methods have lower retain loss at the end of training compared to the "no filter" baseline, reflecting more compute budget spent on non-biology data. SGTM ends with slightly higher retain loss, corresponding to about a 6% compute penalty (Appendix C).

Figure 1 (right) shows the trade-off between the performance on the forget set and on the biology-adjacent knowledge. Here, SGTM also outperforms both data filtering approaches by showing stronger forgetting at any fixed retain loss. The weak filter shows a better trade-off than the strict filter, as the latter disproportionately affects the retain set by removing relevant data categories from training. As a clear example of robustness to label noise, the final retain loss for SGTM lies between weak and strong data filters, while SGTM's forget loss is higher than both weak and strong filters. This shows that SGTM retained some non-biology knowledge from Medicine/Chemistry/Environment domains (removed by the strict filter), while learning less biology from it than the weak filter (which did not filter out these).

It is worth noting that we would expect the loss on biology-adjacent domains to be higher as a result of removing biology data from the training – in fact we see that the final retain loss for the weak filter (no biology) is higher than the baseline. Nevertheless, we still aim to have higher biology loss at any fixed retain loss, as it represents a higher level of preserved capabilities which do not help model's performance on biology.

## 5.3 ROBUSTNESS TO FINE-TUNING

To further assess whether SGTM achieves genuine knowledge localization despite label noise – rather than superficial suppression where mislabeled forget data leaks into retain parameters – we evaluate how quickly the forget knowledge can be re-acquired through adversarial fine-tuning. Figure 4 (right) shows our findings. We perform a full-parameter fine-tuning on a 50/50 mixture of forget and retain data, measuring how quickly the forget knowledge is recovered to the level of the baseline model with no data filtering. Each fine-tuning step represents 260k forget tokens.

Traditional post-training unlearning methods are known to be brittle (Deeb & Roger, 2024; Lermen et al., 2023), which is demonstrated here by the model where biology was unlearned with RMU (Li et al., 2024) recovering the baseline loss within 50 fine-tuning steps (13M forget tokens). Conversely, SGTM shows strong robustness to fine-tuning, requiring the same number of steps – 350 (92M forget tokens) – as the model trained with strict data filtering. Weak filter model took 325 steps (85M forget tokens) to reach the baseline loss.

## 6 DISCUSSION AND LIMITATIONS

**Experimental Limitations.** Our setup differs from real-world deployment in several ways. We use relatively small models (64M and 254M parameters), much smaller than frontier systems. The forget set in our Wikipedia setup ($\sim 4\%$ training tokens) likely exceeds real-world frequencies. We evaluate proxy scenarios (removing Spanish or Wikipedia biology knowledge) rather than genuine CBRN risks, and rely on simpler transformer architectures instead of modern mixture-of-experts. Given computational constraints and needing to train models from scratch, our models are not large enough to yield meaningful results on evaluations that directly probe dangerous capabilities, like WMDP (Li et al., 2024). Our evaluation is thus based on loss metrics as a proxy; this may not reflect downstream task performance or conclusively demonstrate elimination of dangerous capabilities.

**In-Context Attacks.** Shumailov et al. (2024) argues that knowledge removal methods leave models vulnerable to in-context attacks, where adversaries supply dangerous information at inference time. Data filtering has been shown to be susceptible to this (O'Brien et al., 2025), and we expect SGTM to behave similarly, given the nature of our method. However, we view our method as part of a defense-in-depth approach, where knowledge localization and removal serve as one layer among multiple security measures rather than a standalone solution.

**Future Work.** Our method enables the creation of two model versions from a single training run: one possessing dual-use capabilities (before ablation) and another that is safe (after ablation of $\theta_{\text{forget}}$). This dual-model approach offers significant benefits, as trusted actors could benefit from having access to a knowledgeable assistant for legitimate purposes, such as medical research or biosecurity defense. There is evidence suggesting that fine-tuning models on target data post-training may not be as effective as having that data present during pre-training (Kim et al., 2024; Chang et al., 2024). SGTM allows model developers to maintain both versions and deploy them according to their specific use cases and trust requirements. We believe it is an interesting avenue for future work, and we report early findings in Appendix D.

## 7 CONCLUSION

Selective Gradient Masking (SGTM) localizes target knowledge into designated parameters and is robust to label noise. Across both a synthetic bilingual setup and Wikipedia biology removal, SGTM achieves better retain/forget trade-off than data filtering and prior gradient routing variants. In the synthetic setup, we quantify information leakage from undiscovered forget data into non-forget parameters and find it to be minimal. We further show that SGTM is robust to adversarial fine-tuning, in contrast to traditional post-training unlearning methods.

Taken together, these results suggest SGTM as a promising alternative to data filtering, forming part of a broader defense-in-depth strategy for mitigating dual-use capabilities – especially in scenarios where a certain penalty on compute efficiency can be justified by the safety benefits of more reliable capability removal.

ETHICS STATEMENT

This research is motivated by the goal of improving the safe deployment of large language models. The methods we study – particularly Selective Gradient Masking (SGTM) – are designed to reduce the persistence of potentially harmful knowledge in models, thereby benefiting developers and policymakers seeking to mitigate dual-use risks. While any work on knowledge removal carries the possibility of misuse, we believe the contributions of this paper primarily support good-faith efforts to prevent harm and strengthen defense-in-depth strategies for AI safety.

REPRODUCIBILITY STATEMENT

We provide a detailed description of our method in Section 3 and Appendix F. Experimental setups are described in Section 4 and Section 5, with further implementation details in Appendices I and J. We commit to releasing our code upon publication.

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

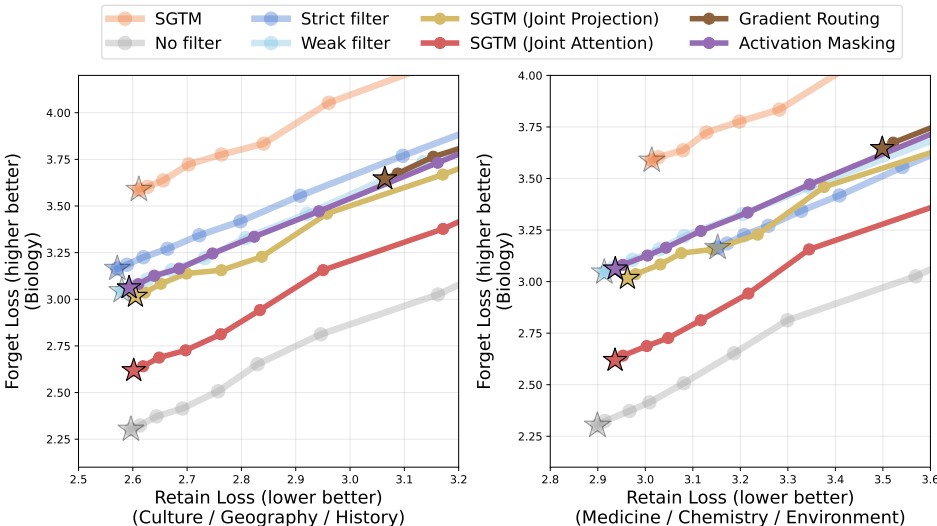

Figure 7: **Retain/forget trade-off when removing target knowledge domain (biology) from a model trained on Wikipedia.** Each line represents the progress of one training run, and each point is a checkpoint at equal intervals of training. Stars show the final checkpoint. Retain loss is computed over general knowledge (left) and biology-adjacent knowledge (right).

## APPENDIX

## A    LLM USE STATEMENT

We used Large Language Models (LLMs) to assist with polishing and smoothing the writing throughout this paper, as well as for coding assistance during implementation. We take full responsibility for all content, ideas, experimental design, results, and conclusions presented in this work.

## B    GRADIENT ROUTING VARIANTS

Gradient Routing (Cloud et al., 2024) was proposed as a generic framework for applying weighted masks over the model's computation graph to isolate the target knowledge into a specified subset of parameters. In this section we explore multiple methods following this framework and show that SGTM, as described in Section 3, provides better retain/forget trade-off than other Gradient Routing variants. Here we use $\theta_{\text{joint}}$ to refer to parameters which are updated by both retain and forget data, but are not removed after training.

Note that below we will use the term Gradient Routing to refer to the particular instantiation of the framework used by Cloud et al. (2024), rather than the framework itself.

In addition to SGTM, we consider the following methods (for details on parameter notation see Appendix F):

- **Gradient Routing** applies zero-masks to the activation gradients. It differs from SGTM in two key aspects. First, the original variant masks the activation gradients, unlike SGTM which masks parameter gradients. Similarly to SGTM that also leads to $\theta_{\text{retain}}$ not being updated on forget examples, but also prevents gradients from being backpropagated, thus changing the gradients for remaining parameters as well – which is disruptive for the model's training. Second, the original variant allows for higher information flow from $\mathbf{D}_{\text{forget}}$ to non-forget parameters: it does not mask all layers (leaving all non-target layers in $\theta_{\text{joint}}$), and by the virtue of masking activation gradients it does not block updates from forget data to down projection layers ($W_2$, $b_2$, $W_O$, $b_O$). For SGTM, on the other hand, $\theta_{\text{joint}}$ is empty by default.

- **Activation Masking** zero-masks $\theta_{\text{retain}}$ during the forward pass, effectively acting as a deterministic data-dependent dropout layer. Similarly to SGTM this strategy also leads to $\theta_{\text{retain}}$ not being updated, but also affects both the loss and the gradients of all parameters. Similar to Gradient Routing, it also does not block updates to down projection layers.

- **SGTM (Joint Projection).** This is a version of SGTM where MLP and attention projection parameters are considered to be joint weights: $\theta_{\text{joint}} = \{W_2, b_2, W_O, b_O\}$

- **SGTM (Joint Attention).** This is a version of SGTM where we only apply masking to the first MLP layer in each transformer block, setting all attention heads and the second MLP layer to be joint weights: $\theta_{\text{joint}} = \{W_2, b_2, W_O, b_O, W_{QKV}^{(i)} \text{ for } i \in \{1, \ldots, h\}\}$. We note that this approach closely resembles Memorization Sinks by Ghosal et al. (2025).

All methods reported in this section use the same masking hyperparameters (if applicable), dedicating $h_{\text{forget}} = 1$ (out of 32) attention head and $d_{\text{forget}} = 64$ (out of 4096) MLP hidden units to the forget data.

Figure 7 shows the retain/forget trade-off on a task of removing biology knowledge from a model trained on Wikipedia (See Section 5). We plot the four methods described above alongside previously reported data for SGTM and three options for data filtering: weak (removing only biology category), strict (also removing biology-adjacent knowledge: Medicine, Chemistry, Environment), and no filtering at all.

Gradient Routing and Activation Masking have trade-off curves similar to the weak filter, but are less compute efficient: for the same compute budget both methods lie further right along the same curve. The compute penalty is most pronounced for Gradient Routing, where the loss at the end of training (both forget and retain) is roughly equivalent to that of a weak filter model at 20% of training.

SGTM variants which do not mask certain gradients (Joint Projection and Joint Attention) do have better final retain loss on both general and biology-adjacent knowledge, but overall provide worse retain/forget trade-off than SGTM, as well as both data filtering options.

## C  COMPUTE PENALTY

Raw loss values can be difficult to interpret due to their non-linear nature. For instance, reducing the loss from 10 to 9 (near random performance) requires substantially less data and compute than reducing from 3 to 2 (where the model already performs well). To provide a more interpretable measure, we convert loss values to the compute budget required to achieve that performance on a baseline model trained with unfiltered data.

### C.1  SCALING LAWS

We establish compute-to-loss scaling laws by training four models of different sizes (34M, 64M, 125M, and 254M parameters) on the Wikipedia dataset. Following Chinchilla scaling principles (Hoffmann et al., 2022), we scale the number of training tokens proportionally with model size while maintaining a constant data mixture. We set number of training tokens to be Chinchilla-optimal in every case (20 × number of parameters).

We compute losses and fit scaling laws separately for three evaluation subsets:

- **Biology** (forget set)

- **Biology-adjacent knowledge** (Medicine/Chemistry/Environment - retain set)

- **General knowledge** (Culture/Geography/History - retain set)

Figure 8 presents the fitted scaling laws. We use the standard approximation of 6 × parameters × tokens to compute FLOPs. The fitted curves follow the form $\ell = \alpha \times C^{-\beta}$ where $\ell$ represents loss and $C$ represents compute in FLOPs.

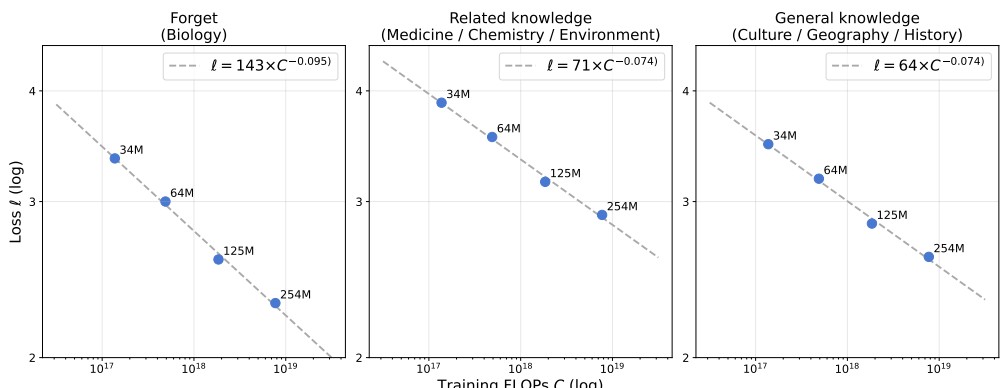

Figure 8: **Scaling laws for models trained on Wikipedia.** We fit and report three compute-to-loss scaling laws, one for each test subset we report: biology (forget), biology-adjacent (retain), general knowledge (retain)

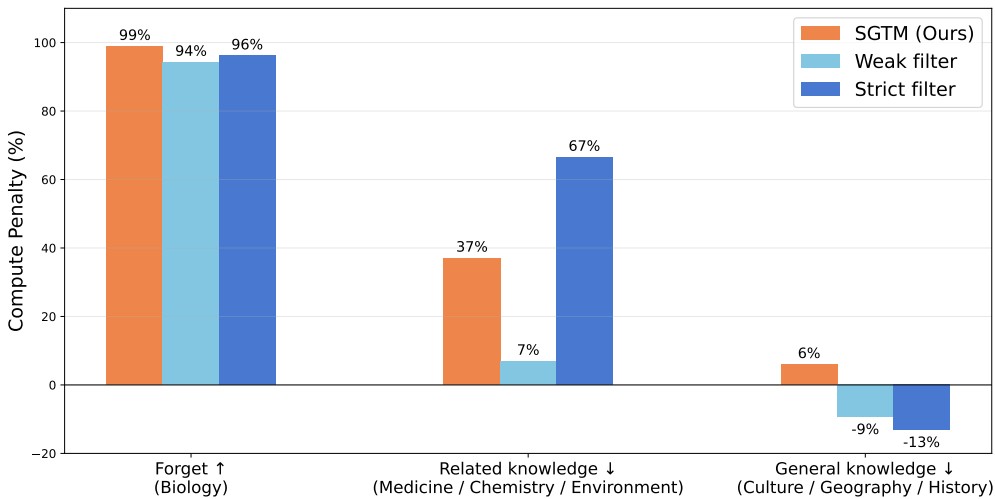

Figure 9: **Compute penalty for three knowledge removal methods.** Reported on three test subsets: biology (forget), biology-adjacent (retain), general knowledge (retain). For each method we convert final loss to the equivalent training compute budget on a baseline (no-filter) model (Figure 8). We then report relative delta between the resulting compute and the baseline.

## C.2    RESULTS

Using these scaling laws, we convert the final losses from each knowledge removal method to equivalent compute budgets. We then report the relative difference compared to baseline training. These values can be interpreted as: "achieving this performance with the baseline model on the original data mix would require X% less compute compared to the full baseline training."

Figure 9 presents the compute penalty analysis across all three evaluation sets.

First, on biology (forget set) both data filtering (94% and 96% penalty) achieve substantial forgetting, with SGTM only slightly outperforming both of them (99% penalty). All methods require less than 6% of the original compute to reach similar biology performance. This high penalty might suggest data filtering could be practical in this setup. However, this reflects limitations of our experimental design: loss metrics serve only as proxies for actual capabilities, and frontier models' high sample efficiency means they could potentially learn from very few examples that escape filtering, making filtering less viable option in practice.

Second, on biology-adjacent knowledge SGTM's 37% penalty falls between weak filtering (7%) and strict filtering (67%). This demonstrates SGTM's key advantage: achieving stronger biology forgetting while preserving some medicine/chemistry/environment knowledge that strict filtering removes entirely. Notably, even weak filtering shows a 7% penalty because removing biology data inherently degrades performance on related domains – an effect that would be more pronounced with perfect filtering.

Finally, on general knowledge SGTM incurs a 6% compute penalty, while both filtering approaches show negative penalties (-9% for weak, -13% for strict). These negative values occur because filtering methods operate at a fixed compute budget, replacing removed biology and biology-adjacent data with additional general knowledge examples. This effectively increases training on general topics beyond what the baseline receives.

The compute penalty analysis reveals important trade-offs between methods. While SGTM requires additional compute compared to data filtering on general tasks, it provides superior forgetting of target knowledge while maintaining better performance on related domains. The 6% penalty on general knowledge represents the computational cost of the gradient masking operations during training – a price that may be justified in certain scenarios by SGTM's robustness advantages demonstrated elsewhere in this work.

## D    PRE-ABLATION PERFORMANCE

In the main paper we evaluate SGTM exclusively after ablation, focusing on label-noise robustness and adversarial fine-tuning resistance. However, selective gradient masking also enables a dual-model deployment strategy: a single training run produces (i) a knowledgeable model with full capabilities, and (ii) a safe variant with target knowledge removed. This paradigm could support workflows where general users receive the safe model, while vetted access is granted to the full-capability version. While this is not the primary goal of our study, we include an initial empirical assessment to illustrate its potential.

Ideally, this setup would satisfy two criteria:

1. **before ablation:** strong performance on both retain and forget domains
2. **after ablation:** strong retention of general knowledge while effectively removing forget-domain capability.

We therefore examine the full trade-off surface rather than focusing solely on post-ablation behavior.

Following Appendix C, we express performance differences in terms of compute penalty relative to the no-filter baseline. Lower values indicate more efficient learning of that capability.

Figure 10 reports compute penalty on three evaluation subsets: forget (Biology), related retain knowledge (Medicine / Chemistry / Environment), and general retain knowledge (Culture / Geography / History). For SGTM variants both pre-ablation (solid) and post-ablation (hatched) values are shown.

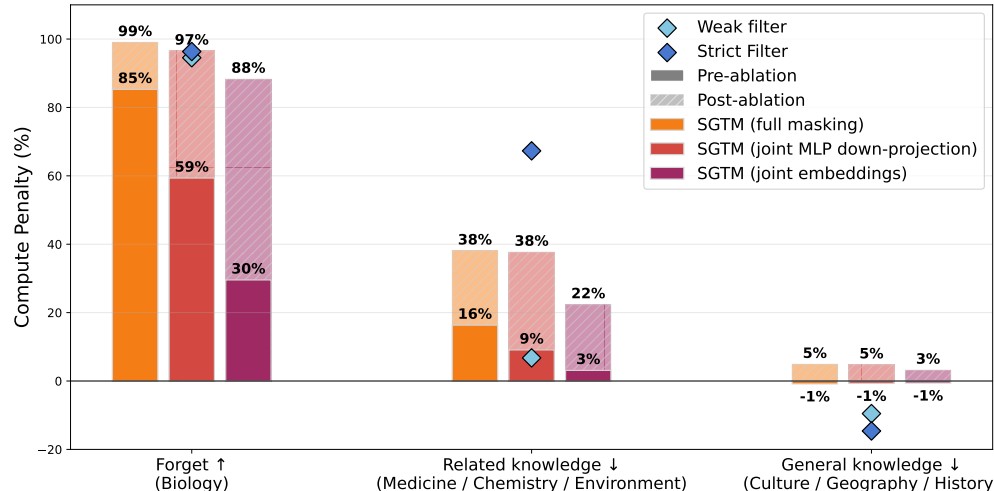

Figure 10: **SGTM can produce a capable pre-ablation model while still enabling strong post-ablation removal.** We report compute penalty relative to a no-filter baseline across forget knowledge (Biology), related retain knowledge (Medicine / Chemistry / Environment), and general retain knowledge (Culture / Geography / History). Solid bars show pre-ablation performance and hatched bars show post-ablation performance. We compare three SGTM variants that differ by how aggressively they mask gradients from biology data: (i) full masking, (ii) joint-projection masking that leaves MLP down-projection parameters unconstrained, and (iii) joint embeddings and down-projections. Weak and strict data-filtering baselines are shown for reference (diamonds). The full masking setup excels post-ablation yet learns biology inefficiently pre-ablation (85% compute penalty). Relaxing gradient masking yields substantially improved biology capability before ablation with minimal retain-set degradation, while still recovering strong forgetting afterward.

We evaluate three SGTM hyperparameter configurations:

- **Default configuration:** $d_{\text{forget}} = 64$; full masking; 10% confident retain sampling. This setup is used for all main-text Wikipedia results.

- **Joint down-projection:** $d_{\text{forget}} = 128$; MLP down-projection is unmasked ($\theta_{\text{joint}} = \{W_2, b_2\}$ as per Appendix F); 10% confident retain sampling.

- **Joint embeddings:** $d_{\text{forget}} = 256$; MLP down-projections and embedding layers are unmasked; 10% confident retain (retaled), 25% confident retain (general).

Weak and strict filtering baselines are included for comparison (diamonds). The no-filter baseline corresponds to zero compute penalty on all subsets.

Figure 10 shows the results. The default configuration – which provides the strongest post-ablation retain/forget trade-off in the main text – also severely underperforms on biology pre-ablation, with a 85% compute penalty. That is, the pre-ablation model never becomes a truly capable "full-access" system on forget knowledge.

Relaxing gradient isolation, however, yields more balanced dual-model behavior. In particular, the joint embeddings configuration achieves substantially better pre-ablation biology performance: 30% compute penalty, with retain-set capabilities nearly unchanged from baseline (3% penalty on related knowledge and 1% improvement on general knowledge. After ablation, it regains strong forgetting with 88% compute penalty on biology, only modestly below data filtering 95%.

These results suggest that SGTM can support dual-model construction when tuned for a more moderate trade-off: the pre-ablation model remains capable on the target domain, while the post-ablation model still removes most of that knowledge. Identifying optimal configurations for such deployments remains an open direction for future work.

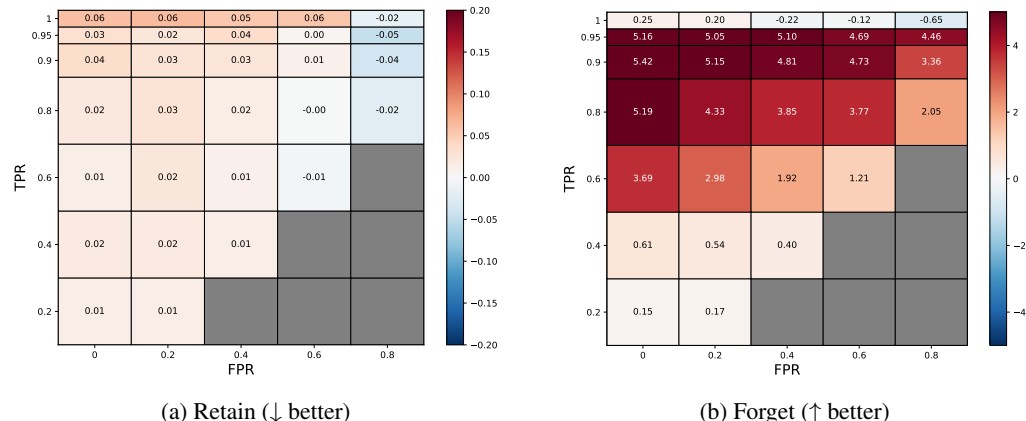

(a) Retain (↓ better)       (b) Forget (↑ better)

Figure 11: **SGTM vs Data Filtering across full input classifier performance range.** We report the difference in loss ($\Delta \ell = \ell_{\text{SGTM}} - \ell_{\text{filter}}$) between SGTM and data filtering on the task of removing the knowledge of Spanish from a model trained on bilingual (English/Spanish) TinyStories dataset. We vary the quality of the input labeling with TPR and FPR of the input classifier.

## E  FULL LABELING SENSITIVITY AND SPECIFICITY ANALYSIS

In Section 4, we examined SGTM's robustness to false negative labeling errors, where forget data is mislabeled as retain. Here we present a comprehensive analysis exploring the full range of classifier performance by varying both false positive rates (FPR) and true positive rates (TPR).

### E.1  EXPERIMENTAL SETUP

We use the same synthetic bilingual TinyStories setup as Section 4, with models trained on English (retain) and Spanish (forget) data. We simulate different classifier performances by artificially controlling TPR and FPR rates. For each (TPR, FPR) pair, we construct the training datasets as follows:

- $\mathbf{D}_{\text{forget}}$: Contains TPR fraction of actual forget data plus FPR fraction of retain data
- $\mathbf{D}_{\text{unlabeled}}$: Contains (1-TPR) fraction of forget data plus (1-FPR) fraction of retain data
- $\mathbf{D}_{\text{retain}}$: Fixed at 10% of ground truth retain data (unchanged across all experiments)

For data filtering, which doesn't train on $\mathbf{D}_{\text{forget}}$, false positives result in less training data overall. All models train for one epoch on their respective datasets, with compute budgets varying based on the amount of filtered data.

### E.2  RESULTS

Figure 11 presents the retain and forget loss differences between SGTM and data filtering ($\Delta \ell = \ell_{\text{SGTM}} - \ell_{\text{filter}}$) across the full spectrum of TPR and FPR values. This visualization captures both classifier quality and decision threshold choice – an ROC curve for any specific classifier can be overlaid on these heatmaps, with each point on the curve representing a different threshold setting. Note that the scales differ between the two subfigures to better visualize the patterns in each metric.

First, Figure 11(a) shows that SGTM consistently shows higher retain loss than data filtering across most of the parameter space, confirming that SGTM incurs a certain penalty in compute efficiency. With the penalty being most pronounced in the top-left region (high TPR, low FPR), the retain performance penalty is decreasing with the decrease in classifier performance.

Second, Figure 11(b) shows that outside of perfect recall (TPR=1), SGTM demonstrates a clear advantage over data filtering, achieving substantially higher forget loss across most classifier operating points. The advantage is particularly strong in regions with moderate TPR (0.4-0.8), where data filtering begins to fail due to leaked forget examples in the training data.

The results suggest that SGTM is particularly valuable when perfect classification is unattainable—a realistic scenario given the difficulty of identifying harmful content at scale. While data filtering can match SGTM's forgetting performance only with perfect recall (TPR=1), achieving this in practice would involve a trade-off with removing useful data.

# F   PARAMETER SPLIT

We consider a transformer block (Vaswani et al., 2017) consisting of a multi-head attention and an MLP layer, with $h$ attention heads, model dimension $d$, MLP dimension $d_{\text{MLP}}$, and per-head dimension $d_h = d/h$.

The trainable parameters in a multi-head attention block are[1]:

$$W_{QKV}^{(i)} \in \mathbb{R}^{3 \times d \times d_h} \qquad \text{(query, key, value for attention head } i\text{)}$$

$$W_O \in \mathbb{R}^{d \times d}, \quad b_O \in \mathbb{R}^d \qquad \text{(output attention projection)}$$

$$W_1 \in \mathbb{R}^{d \times d_{\text{MLP}}}, \quad b_1 \in \mathbb{R}^{d_{\text{MLP}}} \qquad \text{(First MLP layer)}$$

$$W_2 \in \mathbb{R}^{d_{\text{MLP}} \times d}, \quad b_2 \in \mathbb{R}^d \qquad \text{(Second MLP layer)}$$

We designate $h_{\text{forget}}$ attention heads and $d_{\text{forget}}$ MLP hidden units to be dedicated to the forget data, and the remaining attention heads and MLP units to the retain data. We splits weights and biases of the relevant linear layers into forget and retain segments.

We can now define non-overlapping sets of forget ($\theta_{\text{forget}}$) and retain ($\theta_{\text{retain}}$) parameters.

$$W_1 = [W_1^{\text{forget}} \ W_1^{\text{retain}}]; \quad W_1^{\text{forget}} \in \mathbb{R}^{d \times d_{\text{forget}}}, W_1^{\text{retain}} \in \mathbb{R}^{d \times (d_{\text{MLP}} - d_{\text{forget}})}$$

$$W_2 = \begin{bmatrix} W_2^{\text{forget}} \\ W_2^{\text{retain}} \end{bmatrix}; \quad W_2^{\text{forget}} \in \mathbb{R}^{d_{\text{forget}} \times d}, W_2^{\text{retain}} \in \mathbb{R}^{(d_{\text{MLP}} - d_{\text{forget}}) \times d}$$

$$W_O = \begin{bmatrix} W_O^{\text{forget}} \\ W_O^{\text{retain}} \end{bmatrix}; \quad W_O^{\text{forget}} \in \mathbb{R}^{(d_h * h_{\text{forget}}) \times d}, W_2^{\text{retain}} \in \mathbb{R}^{(d_h * (h - h_{\text{forget}})) \times d}$$

$$b_1 = [b_1^{\text{forget}} \ b_1^{\text{retain}}]; \quad b_1^{\text{forget}} \in \mathbb{R}^{d_{\text{forget}}}, b_1^{\text{retain}} \in \mathbb{R}^{(d_{\text{MLP}} - d_{\text{forget}})}$$

$$\theta_{\text{forget}} = \{W_1^{\text{forget}}, b_1^{\text{forget}}, W_2^{\text{forget}}, W_O^{\text{forget}}, W_{QKV}^{(i)} \text{ for } i \in \{1, \ldots, h_{\text{forget}}\}\}$$

$$\theta_{\text{retain}} = \{W_1^{\text{retain}}, b_1^{\text{retain}}, W_2^{\text{retain}}, b_2, W_O^{\text{retain}}, b_O, W_{QKV}^{(i)} \text{ for } i \in \{h_{\text{forget}} + 1, \ldots, h\}\}$$

# G   LEAKAGE DEFINITION

Following the notation from Section 3.1 and Appendix F, we consider SGTM training process $\mathcal{A}_{\text{SGTM}}$ with the dataset split into three subsets: $\mathbf{D}_{\text{SGTM}} = \{\mathbf{D}_{\text{forget}}, \mathbf{D}_{\text{retain}}, \mathbf{D}_{\text{unlabeled}}\}$. We denote subsets of $\mathbf{D}_{\text{unlabeled}}$ with corresponding ground truth labels as $\mathbf{D}_{\text{unlabeled}}^{\text{retain}}$ and $\mathbf{D}_{\text{unlabeled}}^{\text{forget}}$ respectively:

---

[1]For notation simplicity we omit trainable parameters in normalization layers.

$$\theta_{\text{SGTM}} \leftarrow \mathcal{A}_{\text{SGTM}}(\mathbf{D}_{\text{forget}}, \mathbf{D}_{\text{retain}}, \mathbf{D}_{\text{unlabeled}})$$

$$\mathbf{D}_{\text{unlabeled}} = \mathbf{D}_{\text{unlabeled}}^{\text{retain}} \quad \cup \quad \mathbf{D}_{\text{unlabeled}}^{\text{forget}}$$

$$\mathbf{D}_{\text{forget}}, \ \mathbf{D}_{\text{unlabeled}}^{\text{forget}} \sim \mathcal{D}_{\text{forget}}$$

$$\mathbf{D}_{\text{retain}}, \ \mathbf{D}_{\text{unlabeled}}^{\text{retain}} \sim \mathcal{D}_{\text{retain}}$$

Similarly, we consider standard training process $\mathcal{A}_{\text{standard}}$ on a dataset $\mathbf{D}_{\text{standard}}$. While standard training does not distinguish between forget and retain samples during training, we denote parts of $\mathbf{D}_{\text{standard}}$ coming from respective data distributions as $\mathbf{D}_{\text{standard}}^{\text{retain}}$ and $\mathbf{D}_{\text{standard}}^{\text{forget}}$:

$$\theta_{\text{standard}} \leftarrow \mathcal{A}_{\text{standard}}(\mathbf{D}_{\text{standard}})$$

$$\mathbf{D}_{\text{standard}} = \mathbf{D}_{\text{standard}}^{\text{retain}} \quad \cup \quad \mathbf{D}_{\text{standard}}^{\text{forget}}$$

$$\mathbf{D}_{\text{standard}}^{\text{forget}} \sim \mathcal{D}_{\text{forget}}$$

$$\mathbf{D}_{\text{standard}}^{\text{retain}} \sim \mathcal{D}_{\text{retain}}$$

For any given SGTM run we then consider corresponding standard training run with the same retain training data, while varying the size of the forget training data to achieve the same forget loss $\ell(\theta, \mathcal{D}_{\text{forget}})$ as the SGTM run. Below $\mathbf{D}_{\text{forget}}, \mathbf{D}_{\text{retain}}, \mathbf{D}_{\text{unlabeled}}^{*}$ are used to refer to SGTM training data, while $\mathbf{D}_{\text{standard}}^{*}$ refers to standard training run data, and $|\mathbf{D}_{*}|$ refers to the number of samples is a given dataset.

**Definition G.1.** *Leakage* of an SGTM run is then defined as the size of the forget dataset used in standard training required to achieve the equivalent forget loss, normalized by the number of unlabeled forget samples seen by the SGTM run.

$$\textit{Leakage}(\theta_{\text{SGTM}}) := \frac{\left| \mathbf{D}_{\text{standard}}^{\text{forget}} \right|}{\left| \mathbf{D}_{\text{unlabeled}}^{\text{forget}} \right|} \tag{1}$$

Such that

$$\theta_{\text{SGTM}} \leftarrow \mathcal{A}_{\text{SGTM}}(\mathbf{D}_{\text{forget}}, \ \mathbf{D}_{\text{retain}}, \ \mathbf{D}_{\text{unlabeled}}^{\text{retain}} \cup \mathbf{D}_{\text{unlabeled}}^{\text{forget}})$$

$$\theta_{\text{standard}} \leftarrow \mathcal{A}_{\text{standard}}(\mathbf{D}_{\text{standard}}^{\text{retain}} \cup \mathbf{D}_{\text{standard}}^{\text{forget}})$$

$$\mathbf{D}_{\text{standard}}^{\text{retain}} = \mathbf{D}_{\text{forget}} \cup \mathbf{D}_{\text{unlabeled}}^{\text{retain}} \qquad \text{(Same retain data)}$$

$$\ell(\theta_{\text{SGTM}}, \mathcal{D}_{\text{forget}}) = \ell(\theta_{\text{standard}}, \mathcal{D}_{\text{forget}}) \qquad \text{(Same forget loss)}$$

In practice, finding the exact $\mathbf{D}_{\text{standard}}^{\text{forget}}$ so that losses match exactly is computationally intractable. We instead compute losses for a range of increasingly large $\mathbf{D}_{\text{standard}}^{\text{forget}}$ and then perform a linear interpolation between the two closest baseline models.

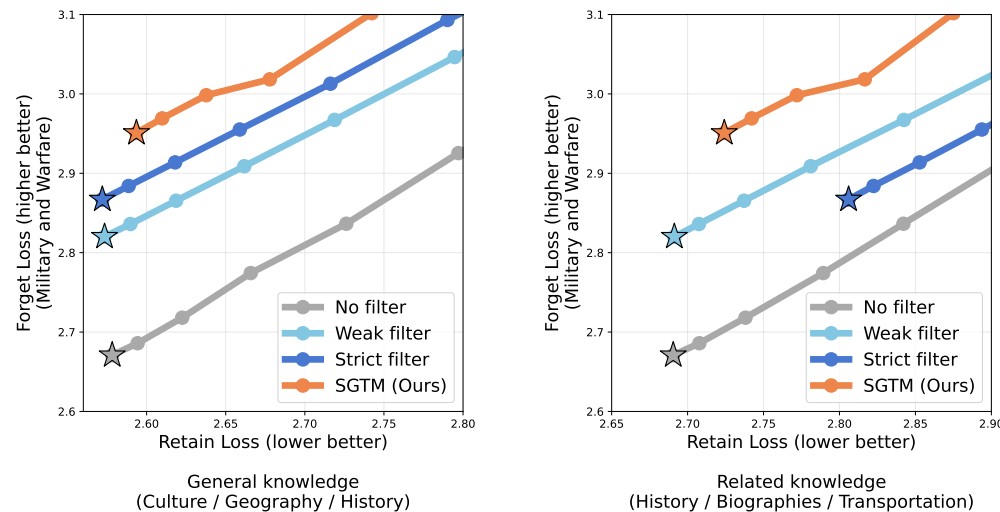

Figure 12: **Retain/forget trade-off when removing "Military and Warfare" category knowledge from a model trained on Wikipedia.**

# H   ADDITIONAL FORGET CATEGORIES

To assess the robustness of our findings beyond the biology domain, we repeat the Wikipedia experiment from Section 5 using an alternative forget category. Instead of removing biology knowledge, we now designate "Military and Warfare" as the forget category. As before, we evaluate performance on (i) the target domain, (ii) related categories (History, Biography, Transportation), and (iii) general knowledge.

Figure 12 shows that the resulting trends closely mirror those observed in our biology experiments. SGTM consistently achieves a better retain/forget trade-off than both weak and strict filtering: at any fixed retain loss, SGTM reaches higher loss on the Military/Warfare forget set. As in Section 5, SGTM ends with higher retain loss, reflecting incurred compute efficiency penalty – but provides stronger forgetting than either filtering strategy.

# I   TINYSTORIES EXPERIMENTAL DETAILS

## I.1   TRAINING HYPERPARAMETERS

All relevant training hyperparameters are listed in Table 2. We aim to roughly double the size of our largest model compared to the model from Eldan & Li (2023) (64M vs. 33M) to account for the extra Spanish data.

## I.2   LOGIT CALIBRATION

To avoid artificially low token probabilities after model ablation, potentially leading to unreasonably high loss values, we perform port-training logit calibration. We train a simple logit bias layer (one parameter per logit, 50257 in total) minimizing the combination of the forget and retain loss: $\ell = \ell_{\text{forget}} + \alpha\ell_{\text{retain}}$. It's important to note that we perform the calibration assuming full access to both forget and retain data, including setups with the full data filtering, where none of the forget data was used during training.

The goal of this step is to stabilise the forget loss post-ablation and recover loss spikes caused by extremely low probabilities (potentially worse than random) of the relevant tokens. As such we set $\alpha = 100$ to make sure we recover the forget loss where possible, but not at the expense of the retain loss.

| Hyperparameter | 64M | 34M | 18M | 8M |
|---|---|---|---|---|
| Parameters | 63,855k | 33,732k | 17,781k | 7,736k |
| Layers | 12 | 8 | 6 | 6 |
| Model dimenstion $d$ | 512 | 384 | 256 | 128 |
| MLP dimension $d_{\text{MLP}}$ | 2048 | 1536 | 1024 | 512 |
| Warmup steps | 1000 | 1000 | 1000 | 500 |
| Training steps | 33120 | 17484 | 9204 | 3989 |
| Vocabulary size | 50257 | | | |
| Attention heads $h$ | 32 | | | |
| Learning rate | 5e-3 | | | |
| Tied embeddings | True | | | |
| LR Schedule | Cosine annealing with warmup | | | |
| Batch size | 128 | | | |
| Context size | 512 | | | |
| Tokenizer | gpt-2 | | | |
| Optimizer | AdamW | | | |
| Weight decay | 0.1 | | | |
| $\beta_1$ | 0.9 | | | |
| $\beta_2$ | 0.95 | | | |

Table 2: Training hyperparameters for synthetic bilingual experiments.

### I.3 DATASET

We use the original TinyStories dataset (Eldan & Li, 2023) in English (466M tokens), and create a Spanish translation of the entire dataset 718M tokens. Note that the Spanish dataset contains more tokens than the English version because the GPT-2 tokenizer was primarily trained on English text, making it less efficient for encoding Spanish. Our final bilingual training dataset consists of both the original English stories and their Spanish translations, totaling 1.2B tokens. In both English and Spanish and additional 1% of data is set aside as a test set (12M tokens).

We consider each story to be an individual training example, truncating and padding the text when necessary.

We translate the Tinystories dataset (Eldan & Li, 2023) into Spanish using `claude-3-haiku-20240307` with the following prompt:

```
Translate the following short story into Spanish.  Keep the
same tone, style, and meaning.  The translation should be
natural and fluent Spanish, appropriate for children.

English story:   story

Spanish translation:
```

Below is an example of an original story in English:

> There was a little girl with dark hair.  Her name was Joy.
> She lived in a big house with her parents.  One day, Joy was
> playing outside in her garden.  Suddenly, she felt something
> on her leg – something pinching her.  It was a big, black bug!
>
> Joy screamed and tried to get away, but the bug kept following
> her.  She tried to run and hide, but it was too quick.
>
> Joy's parents heard her cries and came running.  They used a
> stick to help her get rid of the bug.  After the bug was gone,
> they hugged Joy and told her everything would be alright.
>
> When the bug was gone, Joy felt relieved and happy.  She went
> back to playing in the garden, making sure she didn't step on
> any more bugs.

And the corresponding Spanish version:

> Había una niña pequeña con el cabello oscuro.  Su nombre era
> Alegría.  Vivía en una casa grande con sus padres.  Un día,
> Alegría estaba jugando afuera en su jardín.  De repente,
> sintió algo en su pierna – algo que le estaba pellizcando.
> ¡Era un bicho grande y negro!
>
> Alegría gritó e intentó alejarse, pero el bicho seguía
> persiguiéndola.  Trató de correr y esconderse, pero era
> demasiado rápido.
>
> Los padres de Alegría escucharon sus gritos y corrieron hacia
> ella.  Usaron un palo para ayudarla a deshacerse del bicho.
> Después de que el bicho se fue, abrazaron a Alegría y le
> dijeron que todo estaría bien.
>
> Cuando el bicho se fue, Alegría se sintió aliviada y feliz.
> Volvió a jugar en el jardín, asegurándose de no pisar más
> bichos.

## J   WIKIPEDIA EXPERIMENTAL DETAILS

### J.1   TRAINING HYPERPARAMETERS

All relevant training hyperparameters are listed in Table 3.

For logit calibration we follow the procedure identical to the one described for the TinyStories setup (Appendix I.2).

### J.2   RMU

Using the implementation provided by Li et al. (2024), we apply RMU to the baseline model trained on the full dataset with no filtering. Hyperparameters are reported in Table 4.

## K   ARTICLETOPIC TAXONOMY

| Hyperparameter | 254M | 125M | 64M | 34M |
|---|---|---|---|---|
| Parameters | 254,054k | 124,462k | 64,117k | 33,929k |
| Layers | 16 | 12 | 12 | 8 |
| Model dimenstion $d$ | 1024 | 768 | 512 | 384 |
| MLP dimension $d_{\text{MLP}}$ | 4096 | 3072 | 2048 | 1536 |
| Warmup steps | 1000 | 1000 | 500 | 500 |
| Batch size | 512 | 256 | 256 | 128 |
| Training steps | 9689 | 9491 | 4887 | 5169 |
| Learning rate | 6e-3 | | | |
| LR Schedule | Cosine annealing with warmup | | | |
| Tied embeddings | True | | | |
| Attention heads $h$ | 32 | | | |
| Vocabulary size | 50257 | | | |
| Context size | 1024 | | | |
| Tokenizer | gpt-2 | | | |
| Optimizer | AdamW | | | |
| Weight decay | 0.1 | | | |
| $\beta_1$ | 0.9 | | | |
| $\beta_2$ | 0.95 | | | |

Table 3: Training hyperparameters for Wikipedia model.

| Hyperparameter | Value |
|---|---|
| Steps | 250 |
| $\alpha$ | 100 |
| Steering coefficient | 20 |
| Batch size | 4 |
| Layer to unlearn | 7 |
| Update layers | 5, 6, 7 |
| Update parameters | MLP weights and biases ($W_1$, $b_1$, $W_2$, $b_2$) |

Table 4: RMU hyperparameters.

| Culture | Geography | History and Society | STEM |
|---|---|---|---|
| Biography | Geographical | Business and economics | STEM* |
|    Biography* | Regions | Education | Biology |
|    Women* |   Africa | History | Chemistry |
| Food and drink |     Africa* | Military and warfare | Computing |
| Internet culture |     Central Africa | Politics and government | Earth and environment |
| Linguistics |     Eastern Africa | Society | Engineering |
| Literature |     Northern Africa | Transportation | Libraries and Information |
| Media |     Southern Africa | | Mathematics |
|    Media* |     Western Africa | | Medicine and Health |
|    Books |   Americas | | Physics |
|    Entertainment |     Central America | | Space |
|    Films |     North America | | Technology |
|    Music |     South America | | |
|    Radio |   Asia | | |
|    Software |     Asia* | | |
|    Television |     Central Asia | | |
|    Video games |     East Asia | | |
| Performing arts |     North Asia | | |
| Philosophy and religion |     South Asia | | |
| Sports |     Southeast Asia | | |
| Visual arts |     West Asia | | |
|    Visual arts* |   Europe | | |
|    Comics and Anime |     Eastern Europe | | |
|    Fashion |     Northern Europe | | |
| |     Southern Europe | | |
| |     Western Europe | | |
| |   Oceania | | |

Table 5: Wikipedia `articletopic` taxonomy