# OpenReview forum: "Beyond Data Filtering: Knowledge Localization for Capability Removal in LLMs"
_ICLR.cc/2026/Conference — Submitted to ICLR 2026_

### Official Review · Reviewer_K9fW · 2025-10-21

**Soundness:** 3
**Presentation:** 4
**Contribution:** 3
**Rating:** 8
**Confidence:** 3

**Summary:**

This paper proposes Selective GradienT Masking (SGTM) (an improved version of gradient routing) to remove undesirable knowledge while maintaining desirable performance for LLM pretraining. This approach works by performing:

- Partitioning the model parameters into retain and forget sets;
- During the backward pass, selectively masking out retain gradients when the data are sourced from the forget dataset;
- During the forward pass, selectively masking out forget parameters when the data are sourced from the retain dataset;

The authors evaluate their approach on two datasets: a synthetic bilingual TinyStories case wherein the English version is the retain set and the Spanish version is the forget set, and Wikipedia, wherein the biology knowledge is the forget set. Empirical results show that SGTM consistently outperform gradient routing on both setups, and has the potential to close the gap with data filtering. Additionally, models trained with SGTM are more robust to adversarial finetuning.

**Strengths:**

- The paper is very well-written. I must admit that I'm not an expert in pretraining approaches to unlearning, but the authors have demonstrated expertise in their exposition, overview, and contributions. The literature review also gives a concrete sense of SOTA approaches.
- The proposed SGTM approach is simple and intuitive, making them suitable for large-scale training runs for frontier models.
- The experiments are well-designed, with a more controlled, stylized setting of bilingual TinyStories as a proof of concept, and a more realistic Wikipedia pretraining as validation.
- The authors demonstrate that SGTM is more robust to unlabeled forget data Figure 3(b), albeit at the cost of higher retain loss on identically sized models. One can argue that the effective parameter count is lower due to the masking schemes, and the authors conduct targeted scaling law analysis in the appendix to investigate this.
- It is somewhat expected (but interesting!) that SGTM models are more robust to adversarial finetuning

**Weaknesses:**

- Scale remains a primary concern, as the largest models studied in this paper is 254M. While I don't deduct points from it due to the expense of pretraining, whether this approach can be scaled to frontier systems is an open question, and whether the compute penalty (6% for general knowledge according to the authors) is a worthy trade-off remains to be studied.

**Questions:**

- I really like the masking idea, but I'm curious if the authors should explore masking in the output space, i.e. by masking out the loss of undesirable tokens so that the model can have a good understanding of the surrounding context, but at inference time they will not generate these undesirable knowledge.

---

> ### Author Response · Authors · 2025-11-20
>
> We thank the reviewer for the encouraging and thoughtful feedback. We appreciate that they found the two-stage experimental design compelling: the controlled TinyStories setup and the realistic Wikipedia evaluation were crafted precisely to provide both interpretability and real-world relevance, and we are glad that this structure resonated with the reviewer. We are also pleased the reviewer has highlighted our adversarial fine-tuning results, as we believe that it is one of the more compelling points supporting the robustness of our method.
>
> Response to questions and weaknesses:
>
> > Scale remains a primary concern, as the largest models studied in this paper is 254M.
>
> We agree that the scale of our experiments is a limitation, and we appreciate the reviewer recognizing the pretraining costs as a limiting factor.
>
> We, however, make an attempt at understanding how our method scales to larger models. We have conducted **additional experiments to estimate how performance changes with model and dataset size**. In particular, in Section 4.3 – where we quantify information leakage from forget data into retain parameters – we expanded the analysis from a single model to a series of models ranging from 8M to 64M parameters (with training datasets scaled accordingly). Figure 5(b) shows that **leakage decreases with scale**, indicating that larger models exhibit better knowledge localization and stronger robustness to labeling errors.
>
> While these experiments still involve only relatively small models and do not replace large-scale evaluation, we believe they provide reasonably strong evidence that our method is likely to remain effective at a larger scale.
>
> > I really like the masking idea, but I'm curious if the authors should explore masking in the output space, i.e. by masking out the loss of undesirable tokens so that the model can have a good understanding of the surrounding context, but at inference time they will not generate these undesirable knowledge.
>
> This is an interesting discussion point, and we thank the reviewer for raising this. The proposed masking idea, if we understand it correctly, looks similar to the method proposed by Wang et al. (2025), highlighted by Reviewer w7AC – where the model is trained to understand but not generate harmful content. This is a promising direction in its own right and can be more appropriate in scenarios where retaining underlying understanding is desirable (e.g., toxic or copyright-sensitive material). However, it leads to a notably different trade-off and type of model behavior compared to our goal: SGTM aims to remove all traces of the target knowledge so that it cannot be recovered even under adversarial fine-tuning, whereas methods in this family explicitly maintain the model’s understanding while suppressing generation.
>
> ---
>
> References:
>
> Wang, R., Finlayson, M., Soldaini, L., Swayamdipta, S., & Jia, R. (2025). Teaching Models to Understand (but not Generate) High-risk Data. arXiv preprint arXiv:2505.03052.

---

### Official Review · Reviewer_FJtB · 2025-10-31

**Soundness:** 4
**Presentation:** 4
**Contribution:** 4
**Rating:** 6
**Confidence:** 3

**Summary:**

This paper proposes a method for unlearning in a LLM at the pretraining stage, by masking gradients to a parameter set on the backward pass if an example is part of the forget set. They argue this is more robust to label noise than data filtering or the similar Gradient Routing. They support this with empirical evidence.

**Strengths:**

- clearly written, seems novel - caveat that I am not deeply familiar with the unlearning literature
- experiments seem to support the basic point of improvement the authors suggest for SGTM, and are fairly thorough (ablations with related data categories are cool)
- Fig 1 is great! In general the communication around tradeoffs is well done

**Weaknesses:**

- it's odd to me that there aren't results shown for Fig 4 for GR - isn't this the main baseline we should be comparing to?
- some contradictory statements around parameter subsets: in Fig 2 caption the authors that the after forget parameters are assigned, “the remaining parameters are designated to the retain data” but then discuss something called "joint" parameters in line 183
- it would be good to give more intuition here - why is SGTM more robust to label noise? it's not a priori obvious to me that it should be, some exploration about the difference to baselines would be helpful
- I'm not sure exactly how this is usually handled in unlearning, but there doesn't seem to be a lot of information about how the SGTM model performs without the parameters masked
- as someone not deeply familiar with the literature, it's not quite clear to me if the only difference between SGTM and GR is activation vs. parameter gradient masking? would be good to state this more clearly
- Leakage: defining this as a percentage is odd to me - it's misleading that the number is constant between 5% and 50% tokens, since that's the equivalent of 4x tokens, which is a lot! I think something that scales with the token equivalence number (eg 707k in Fig 5a) is more sensible

**Questions:**

- how does GR perform on the real data in Fig 4?
- why does SGTM display more robustness to label noise?
- clarify: main difference between GR and SGTM is activation vs parameter gradient masking?

---

> ### Author Response · Authors · 2025-11-20
>
> We thank the reviewer for encouraging feedback and thoughtful questions, which we believe have made the paper stronger.
>
> Response to questions and weaknesses:
>
> > how does GR perform on the real data in Fig 4?
>
> We appreciate the reviewer’s suggestion.
>
> **We have updated Figure 4 (left) to include GR in the revised version of the paper**. In hindsight, we recognize that excluding Gradient Routing from Figure 4 was not the best presentation choice. Our original reasoning was that GR performs strictly worse than SGTM, showing both higher retain loss and lower forget loss, and therefore including GR in the trade-off plot was unnecessary. We, however, completely agree that showing GR explicitly improves completeness and clarity.
>
> For the reviewer’s reference, in addition to TinyStories results on Figure 4, we also report data for GR for our Wikipedia experiments in Appendix B (Figure 7), where it also underperforms compared to SGTM.
>
> > why does SGTM display more robustness to label noise?
>
> We appreciate the reviewer’s question, and we are excited to unpack the intuition behind SGTM’s robustness to label noise.
>
> We have introduced a **new experiment** measuring per-sample gradient norms for unlabeled forget and retain examples (i.e., applying no masking). Figure 6 (Section 4.4) shows that **unlabeled forget data predominantly updates forget parameters, and unlabeled retain data predominantly updates retain parameters**. This shows that the model develops self-reinforcing knowledge localization: once labeled forget examples begin steering updates into forget weights, unlabeled forget samples naturally follow the same pathway and send stronger gradient signals to forget parameters even without explicit masking.
>
> > clarify: main difference between GR and SGTM is activation vs parameter gradient masking?
>
> We appreciate this question from the reviewer, and we agree that the submitted version of the paper does not explain the differences clearly.
>
> We have now **updated Section 3.2 to highlight the differences between the two methods**, and refer more explicitly to the Appendix B where we elaborate on the differences.
>
> The key difference between the two methods is, indeed, as the reviewer points out, that Gradient Routing masks activation gradients, while SGTM masks parameter gradients.
>
> To quote from the Appendix B:
>
> Similarly to SGTM, GR's approach leads to $\theta_\text{retain}$ not being updated on forget examples, but also prevents gradients from being backpropagated, thus changing the gradients for remaining parameters as well -- which is disruptive for the model's training. The original variant also allows for higher information flow from forget data to non-forget parameters: it does not mask all layers, and by the virtue of masking activation gradients it does not block updates from forget data to down projection layers.
>
> > some contradictory statements around parameter subsets
>
> We apologize for the confusion caused by the original version of the notation.
>
> In the standard version of our approach, $\theta_{\text{joint}}$​ is empty and is not required to describe the core algorithm. To avoid this confusion, we have removed $\theta_{\text{joint}}$ from Section 3.1 (“Notation”) and now introduce it only in Appendix B, where we examine alternative gradient-masking configurations that include non-empty joint parameter sets.
>
> We initially introduced it alongside $\theta_{\text{retain}}$​ and $\theta_{\text{forget}}$​ for use in later exploratory analyses, but we now recognize that mentioning it in the main notation section created unnecessary ambiguity.
>
> > but there doesn't seem to be a lot of information about how the SGTM model performs without the parameters masked
>
> We think this is a very interesting question to explore, and sincerely thank the reviewer for raising this. While it is not the main focus of this paper, we briefly discuss in Section 6 that **SGTM enables the  creation of two model versions from a single training run**: one possessing dual-use capabilities (before ablation) and another that is safe (after ablation forget weights).
>
> **We have conducted an exploratory analysis of the pre-ablation model and report the results in Appendix D**. SGTM’s strong gradient isolation causes the pre-ablation model to perform noticeably worse on the forget domain. However, when we relax gradient isolation, allowing more information to flow into the retain parameters, the pre-ablation model recovers much stronger performance. This improvement comes with a clear trade-off: forgetting becomes weaker after ablation. Nonetheless, this variant demonstrates that SGTM can be tuned along a spectrum, supporting either stronger capability removal or more balanced dual-model behavior, depending on the use case.
>
> ---
>
> We hope that our revisions and additional experiments adequately address the reviewer’s
> concerns. If the reviewer agrees, we kindly ask them to consider updating their assessment of the paper.

---

### Official Review · Reviewer_w7AC · 2025-11-01

**Soundness:** 3
**Presentation:** 3
**Contribution:** 2
**Rating:** 4
**Confidence:** 4

**Summary:**

Undesirable data is often filtered out during pre-training, but filters have false negatives, so some undesirable content remains in the pre-training data. This paper proposes a training technique that encourages such undesirable data to reside in a small subset of parameters that can then be removed.

**Strengths:**

Empirical results on multiple settings show an improved trade-off between general capabilities and forgetting of undesirable content, compared with filtering. Moreover, it has much better performance against fine-tuning compared with a strong unlearning method, RMU.

The method is quite simple and intuitive. Gradient masking sequesters undesirable knowledge into a small subset of parameters, while parameter masking encourages the rest of the parameters to function well even when those parameters are removed.

The paper is well-written and clear.

**Weaknesses:**

This paper compares only with filtering and a similar previous work (Gradient Routing), but other methods have also been developed as alternatives to filtering:
* https://arxiv.org/abs/2302.08582 This paper explores several training objectives and finds that a "conditional training" approach works well. It seems that SGTM could directly compare with this approach.
* https://arxiv.org/abs/2505.03052 This paper has a somewhat different motivation, but they can use a more aggressive threshold on the classifier because their intervention is less strict than filtering. It also seems worth discussing and possibly comparing with

$\theta_{\text{retain}}$ is used but is not clearly defined (e.g. Line 182). It's also confusing that the retain parameters are not mentioned in Lines 250-253.

The experimental settings are somewhat toy. For TinyStories, 64M is a very small model. It is also quite synthetic to generate the Spanish data with translation, when multilingual corpora also exist. The noise model is also not realistic, as it is pure iid noise. Finally, the motivation of this experiment is a bit unclear, since in practice one would not want to prohibit the model from learning a second language. The Wikipedia experiments are more realistic in noise, though 254M is still quite a small model.

The experiments are also somewhat narrow. Only these two model sizes are considered, and for Wikipedia only one possible forget set is considered. Perhaps toxic text could be considered as another type of data that is typically filtered but only imperfectly.

Finally, the paper does not clearly explain the methodological difference with the previous version of Gradient Routing from Cloud et al. (2024). This is important for explaining the novelty of this method.

**Questions:**

Could you explain more what $\theta_{\text{retain}}$ is, how it differs from $\theta_{\text{joint}}$, and how it is used?

What are the main differences between SGTM and Cloud et al.? From looking at that paper, it seems that the difference may be from the selective parameter masking, but I am curious if this is correct and if there are other differences.

---

> ### Author Response · Authors · 2025-11-20
>
> We thank the reviewer for their careful assessment of our work and appreciate the time taken to provide detailed feedback.
>
> **Response to weaknesses and questions:**
>
> > Could you explain more what $\theta_{\text{retain}}$ is, how it differs from  $\theta_{\text{joint}}$​ , and how it is used?
>
> We thank the reviewer for this question, and we appreciate the opportunity to improve the clarity of the presentation for a key concept in our work.
>
> We now explicitly clarify in the main body of the paper that retain parameters are all parameters not designated as $\theta_{\text{forget}}$, while $\theta_{\text{joint}}$ is empty in the standard version of SGTM. To avoid the confusion, we have removed $\theta_{\text{joint}}$ from Section 3.1 (“Notation”) and now introduce it only in Appendix B, where we examine alternative gradient-masking configurations that include non-empty joint parameter sets. Detailed description of the parameter split is provided in Appendix F.
>
> The way $\theta_{\text{retain}}$ is used is described in Section 3.2 and Table 1. For all forget data samples we zero out gradients for $\theta_{\text{retain}}$ ($\nabla_{\theta_{\text{retain}}}=0$), thus ensuring that forget samples only update forget parameters.
>
> We apologize for the confusion caused by the original version of the notation with $\theta_{\text{joint}}$. We initially introduced it alongside $\theta_{\text{retain}}$​ and $\theta_{\text{forget}}$​ for use in later exploratory analyses, but we now recognize that mentioning it in the main notation section created unnecessary ambiguity.
>
> > What are the main differences between SGTM and Cloud et al.? From looking at that paper, it seems that the difference may be from the selective parameter masking, but I am curious if this is correct and if there are other differences.
>
> We appreciate this question from the reviewer, and we agree that the submitted version of the paper does not explain the differences clearly.
>
> We have now **updated Section 3.2 to highlight the differences** between the two methods, and refer more explicitly to the Appendix B where we elaborate on the distinction.
>
> To quote from the Appendix B:
>
> Gradient Routing <...> differs from SGTM in two key aspects. First, it masks the activation gradients, unlike SGTM which masks parameter gradients. Similarly to SGTM that also leads to $\theta_\text{retain}$ not being updated on forget examples, but also prevents gradients from being backpropagated, thus changing the gradients for remaining parameters as well -- which is disruptive for the model's training. Second, the original variant allows for higher information flow from forget data to non-forget parameters: it does not mask all layers (leaving all non-target layers in $\theta_\text{joint}$), and by the virtue of masking activation gradients it does not block updates from forget data to down projection layers.
>
> > This paper compares only with filtering and a similar previous work (Gradient Routing), but other methods have also been developed as alternatives to filtering:
>
> We thank the reviewer for pointing out these additional approaches. We agree that our positioning relative to the broader alignment literature should be clearer, and **we have expanded the Related Work section to discuss pretraining-time interventions** aimed at reducing harmful generations, including Korbak et al. (2023) and Wang et al. (2025). Both lines of work make valuable contributions to the space of pretraining-time safety interventions, and highlighting these methods helps situate our work within a broader set of promising, complementary approaches.
>
> However, the goals of our method differ substantially from those of these generation-oriented interventions. **Our objective is to prevent the model from acquiring the harmful knowledge** in the first place, with a particular focus on deep removal such that this knowledge cannot be recovered even under adversarial fine-tuning. At the same time generation-oriented interventions focus on reducing or eliminating harmful outputs, not necessarily underlying model knowledge. In particular, Wang et al. (2025) explicitly evaluate and seek to maintain the model’s understanding of the harmful domain: ”*We demonstrate that models trained with SLUNG on documents about fictitious entities can accurately answer questions about these entities while avoiding the generation of entity names.*”
>
> > For TinyStories, 64M is a very small model.
>
> We respectfully disagree that a 64M-parameter model is very small for the TinyStories setting. With ~1.2B tokens in the combined English+Spanish corpus, a **64M-parameter model is close to the Chinchilla-optimal ratio** of roughly 20 training tokens per parameter (Hoffmann et al., 2022). Moreover, the largest model released by Eldan & Li (2023) alongside the TinyStories dataset contains 33M parameters; our choice of 64M approximately doubles this capacity to account for the expanded bilingual dataset.

---

> > ### Author Response · Authors · 2025-11-20
> >
> > > Finally, the motivation of this experiment is a bit unclear, since in practice one would not want to prohibit the model from learning a second language.
> >
> > We thank the reviewer for raising this concern and appreciate the opportunity to clarify the motivation behind our experimental design. **We have updated the introduction to reflect this feedback**.
> >
> > Our experiments are structured in two stages, each serving a distinct purpose. The first stage, the bilingual TinyStories setup, is deliberately synthetic. Here, we have access to perfect ground-truth labels and a clean separation between the retain and forget data. This setup enables a controlled evaluation: the ideal goal of our method is to approximate a model that has never seen the forget data, and having access to an oracle model trained on retain data only allows us to measure how closely SGTM achieves this ideal – something that is impossible to evaluate in real-world datasets due to label noise. This synthetic setup also allows us to quantify leakage, which cannot be measured without ground-truth data labels.
> >
> > The second stage, the Wikipedia biology-removal experiment, serves as our realistic evaluation. Here the label noise is real rather than synthetic, and the task reflects a more realistic use case where such techniques could be practically relevant.
> >
> > While we agree that removing a language is an artificial objective that one would not pursue in practice, we emphasize that the TinyStories experiment is intended solely as a clean testbed to demonstrate the core properties of SGTM: effective knowledge localization and robustness to label noise. The realistic Wikipedia setup then evaluates the method in a domain where capability removal is meaningful.
> >
> > > The Wikipedia experiments are more realistic in noise, though 254M is still quite a small model.
> >
> > We agree with the reviewer that, given our method requires pretraining from scratch, the scale of our experiments remains a limiting factor, which we explicitly acknowledge in Section 6.
> >
> > We now include an additional experiment to evaluate how SGTM behaves as scale increases, showing that its **performance improves as models grow larger**. In Section 4.3 – where we quantify information leakage from forget data into retain parameters – we expanded the analysis from a single model to a series of models ranging from 8M to 64M parameters (with training datasets scaled accordingly). Figure 5(b) shows that leakage decreases with scale, indicating that larger models exhibit better knowledge localization and stronger robustness to labeling errors.
> >
> > While these experiments still involve only relatively small models and do not replace true large-scale evaluation, we believe they provide preliminary evidence that our method is likely to remain effective at a larger scale.
> >
> > > The experiments are also somewhat narrow. Only these two model sizes are considered, and for Wikipedia only one possible forget set is considered. Perhaps toxic text could be considered as another type of data that is typically filtered but only imperfectly.
> >
> > We thank the reviewer for these thoughtful suggestions, which we find both interesting and promising.
> >
> > First, as noted above, **we have expanded the TinyStories experiments from a single model to a suite of four models ranging from 8M to 64M parameters**, and we now report the resulting scaling trends in Section 4.3. We are grateful for this suggestion, as we believe the observed scaling behaviour is one of the most valuable findings of the paper, highlighting the method’s potential for larger models.
> >
> > Second, we agree that applying the method to additional forget sets would further strengthen the contribution and demonstrate robustness across different scenarios. In response, **we have added a new Appendix H in which we apply SGTM to remove knowledge from the “Military and Warfare” category.** The results remain broadly consistent with those reported for the biology domain, supporting the generality of our findings.
> >
> > To elaborate on our original experiment design, we selected the biology domain because of its relevance to CBRN-related risk, which is one of the primary motivations for knowledge-removal techniques. This choice also allows us to study how biology content is entangled with adjacent domains  such as chemistry and medicine. Our intention was to focus on a domain where underlying knowledge, rather than stylistic content, is central.
> >
> > ---
> >
> > We hope that our revisions and additional experiments adequately address the reviewer’s
> > concerns. If the reviewer agrees, we kindly ask them to consider updating their assessment of the paper.
> >
> > ---
> >
> > References
> >
> > Korbak, Tomasz, et al. "Pretraining language models with human preferences." _International Conference on Machine Learning_. PMLR, 2023.
> >
> > Wang, Ryan, et al. "Teaching Models to Understand (but not Generate) High-risk Data." _arXiv preprint arXiv:2505.03052_ (2025).

---

### Official Review · Reviewer_JtvZ · 2025-11-03

**Soundness:** 3
**Presentation:** 3
**Contribution:** 2
**Rating:** 4
**Confidence:** 3

**Summary:**

The paper introduces selective gradient masking, a pretraining time technique to localize and remove specific capabilities from LLMs. The authors evaluates the method on two settings of (1). synthetic bilingual data and (2). wikipedia corpus. Across both, SGTM achieves a better retain/forget tradeoff under label noise compared to the baselines. The authors also show SGTM is more robust to adversarial fine-tuning.

**Strengths:**

1. **Adversarial robustness**: the detailed discussions on mislabeled content and adversarial fine-tuning are valuable and highly relevant to the community.
2. **Clear presentations**: the figures and visualizations are informative and well-designed.

**Weaknesses:**

1. **Insufficient Evidence**: This is my primary concern. The evaluation relies solely on model loss, which may not adequately capture downstream perfromance differences that truly matter. It is unclear to me whether higher loss indeed indicates better forgetting. Including additional evaluations for forgetting and general performance retention would substantially strengthen the paper's empirical support.

2. **Limited Scale**: As noted in section 6, the experiments use very small model and dataset sizes. It remains uncertain whether the findings would generalize to larger models of real-world training scales.

**Questions:**

1. How were the data points in Figure 1 obtained? Do they represent Pareto frontiers or averages?

2. What is $\theta_{joint}$ specifically? How are the $\theta_{joint}$ parameters selected, and how are they different from $\theta_{retain}$ or $\theta_{forget}$?

---

> ### Author Response · Authors · 2025-11-20
>
> We thank the reviewer for their thoughtful feedback and for the time spent evaluating our work. We are glad that they found the adversarial robustness results compelling, as we view robustness under adversarial fine-tuning as particularly strong evidence that our method removes the target knowledge in a durable rather than superficial way.
>
> **Response to questions and weaknesses:**
>
> > How were the data points in Figure 1 obtained? Do they represent Pareto frontiers or averages?
>
> The data points in Figure 1 correspond to intermediate checkpoints, which can be interpreted as models of the same size trained on smaller amounts of data. Each curve represents the trajectory of a single training run, and the plotted values are **averages over three runs**. We have now clarified this in Section 5.1.
>
> > What is $\theta_\text{joint}$ specifically? How are the  parameters selected, and how are they different from $\theta_\text{retain}$ or $\theta_\text{forget}$?
>
> We apologize for the confusion caused by the original version of the notation. In the standard version of our approach, $\theta_{\text{joint}}$​ is empty and is not required to describe the core algorithm. To avoid this confusion, we have removed $\theta_{\text{joint}}$ from Section 3.1 (“Notation”) and now introduce it only in Appendix B, where we examine alternative gradient-masking configurations that include non-empty joint parameter sets.
> We initially introduced it alongside $\theta_{\text{retain}}$​ and $\theta_{\text{forget}}$​ for use in later exploratory analyses, but we now recognize that mentioning it in the main notation section created unnecessary ambiguity.
>
> > The evaluation relies solely on model loss, which may not adequately capture downstream perfromance differences that truly matter.
>
> We agree with the reviewer that evaluating performance directly on a downstream benchmark such as WMDP (Li et al., 2024) would strengthen our empirical claims. However, such evaluations do not produce meaningful results at the scale we operate in this work, given our method requires pretraining models from scratch, with models performing at near-random on multiple-choice biology tasks such as WMDP or MMLU.
>
> However, based on prior works, we believe **loss is a reasonable proxy** for downstream task performance. A substantial body of research (Du et al., 2024; Gadren et al., 2025; Schaeffer et al., 2023) shows a **strong correlation between benchmark accuracy and pretraining loss**. Schaeffer et al. (2023), for instance, show that many purported “emergent” jumps in benchmark performance can be explained by applying nonlinear or discontinuous metrics to models whose per-token loss actually decreases smoothly and predictably with scale; when more continuous metrics are used, task performance also changes smoothly with loss. Taken together, these results suggest that the higher biology loss we observe is likely to translate into lower performance on biology-related benchmarks when scaled to larger models.
>
> > Limited Scale: As noted in section 6, the experiments use very small model and dataset sizes. It remains uncertain whether the findings would generalize to larger models of real-world training scales.
>
> We agree with the reviewer that the scale of our experiments remains a limiting factor, which we explicitly acknowledge in Section 6.
>
> We now include an additional experiment to evaluate how SGTM behaves as scale increases, showing that its **performance improves as models grow larger**. In Section 4.3 – where we quantify information leakage from forget data into retain parameters – we expanded the analysis from a single model to a series of models ranging from 8M to 64M parameters (with training datasets scaled accordingly). Figure 5(b) shows that leakage decreases with scale, indicating that larger models exhibit better knowledge localization and stronger robustness to labeling errors.
>
> While these experiments still involve only relatively small models and do not replace true large-scale evaluation, we believe they provide preliminary evidence that our method is likely to remain effective at a larger scale.
>
> ---
>
> We hope that our revisions and additional experiments adequately address the reviewer’s
> concerns. If the reviewer agrees, we kindly ask them to consider updating their assessment of the paper.
>
> ---
>
> **References**
>
> Du, Zhengxiao, et al. "Understanding emergent abilities of language models from the loss perspective." Advances in neural information processing systems 37 (2024): 53138-53167.
>
> Gadre, Samir Yitzhak, et al. "Language models scale reliably with over-training and on downstream tasks." The Thirteenth International Conference on Learning Representations
>
> Schaeffer, R., Miranda, B., & Koyejo, S. (2023). Are emergent abilities of large language models a mirage?. Advances in neural information processing systems, 36, 55565-55581

---

> > ### Comment · Reviewer_JtvZ · 2025-11-28
> >
> > Thanks for the responses, they address my concerns, particularly the clarification on "*loss is a reasonable proxy for downstream performance*" and the scaling experiment in Section 4.3. Overall, I think this is a good paper. I will raise the score to 6.

---

### Author Response · Authors · 2025-11-20
**Global response**

We thank all reviewers for their thoughtful and constructive feedback. Their suggestions significantly improved the clarity of the paper and motivated several new experiments that strengthened our empirical findings.

Below we highlight the main changes to the paper, focusing on issues that appeared across multiple reviews. Significant changes are highlighted in blue in the revised version of the paper.

**1. Difference with Gradient Routing (Cloud et al., 2024)**

We have elaborated on the difference between the two methods in Section 3.2 and more explicitly reference the Appendix B where we provide detailed comparison.

**2. Scaling experiments**

We partially address reviewers’ concerns regarding the small scale of our experiments by showing that SGTM becomes more robust to label noise as model and dataset size increases (Section 4.3, Fig 5(b)). While these experiments still involve only relatively small models and do not replace large-scale evaluation, we believe they provide preliminary evidence that our method is likely to remain effective at a larger scale.

**3. Method intuition and gradient norm analysis**

To provide clearer intuition for why SGTM is robust to label noise, we added a new experiment (Section 4.4, Fig. 6) measuring per-sample gradient norms while treating all data as unlabeled (i.e. without masking). These results reveal a specialization effect: **unlabeled forget data predominantly updates forget parameters, and unlabeled retain data predominantly updates retain parameters**.

**4. Joint parameters**

We have clarified the notation and have now removed $\theta_{\text{joint}}$ from Section 3.1 (“Notation”), as it remains empty in the standard version of our approach. We now introduce it later in Appendix B, where we examine alternative gradient-masking configurations that include non-empty joint parameter sets.

---

### Meta-Review · Area_Chair_PMf2 · 2025-12-05

**Summary:**

The paper is clear and proposes a plausible pretraining-time localization method, with encouraging signs (better retain/forget trade-offs; robustness to adversarial fine-tuning). However, the empirical evidence remains too limited for acceptance: results are at small scale; downstream task metrics are absent; comparisons to broader pretraining-time mitigations remain thin; and noise/setting realism is still weak despite added clarifications and minor extensions. The novelty over prior Gradient Routing is clearer but not decisively validated at realistic scales. Overall, the contribution is promising yet insufficiently substantiated for ICLR acceptance at this time.

**Reviewer Concerns:**

**JtvZ** Concerns: reliance on loss as proxy for downstream performance; small scale; unclear notation/role of theta_joint. The rebuttal clarified notation, added GR comparisons and a gradient-norm intuition, and presented small scaling trends. Addressed: notation/intuition; partial scaling evidence. Outstanding: no downstream/task-level evaluations; true large-scale results remain absent.

**w7AC** Concerns: missing baselines beyond filtering/GR; toy/unrealistic settings and noise; unclear novelty vs. Cloud et al.; uneven exposition of theta_retain and theta_joint. The rebuttal expanded related work and clarified SGTM vs. GR; added a second forget set. Addressed: clarity/related work; minor breadth. Outstanding: still limited scale; still narrow empirical scope; lack of realistic noise studies and broader baselines.

**FJtB** Concerns: why SGTM is more noise-robust, absence of GR in Fig. 4, contradictions around parameter subsets, leakage metric. The rebuttal added GR to plots, clarified notation, and offered intuition via gradient norms. Addressed: GR comparison, notation, intuition. Outstanding: leakage metric choice and pre-/post-ablation behavior remain lightly explored.

**K9fW** Concerns: scale, as the largest models studied in this paper is 254M. Addressed: none. Outstanding: scale.

**Reviewer Scores:**

**JtvZ** Already indicated a raise to 6 after rebuttal (which cannot be counted due to the leak and possible collusion).

**w7AC** Empirical breadth/scale remain weak, likely no change.

**FJtB** Major external validity gaps persist, likely no change.

**K9fW** May hold at 8, but his/her support doesn't overcome evidence gaps highlighted by other reviewers.

---

### Decision · Program_Chairs · 2026-01-26

Reject